# Benchmarking Intelligent LLM Agents for Conversational Tabular Data Analysis

## Abstract

Conversational Tabular Data Analysis, a collaboration between humans and machines, enables real-time data exploration for informed decision-making. The challenges and costs of collecting realistic conversational logs for tabular data analysis hinder comprehensive quantitative evaluation of Large Language Models (LLMs) in this task. To mitigate this issue, we introduce **Tapilot-Crossing**, a new benchmark to evaluate LLMs on conversational data analysis. Tapilot-Crossing contains 1024 conversations, covering 4 practical scenarios: Normal, Action, Private, and Private Action. Notably, Tapilot-Crossing is constructed by an economical multi-agent environment, **Decision Company**, with few human efforts. This environment ensures efficiency and scalability of generating new conversational data. Our comprehensive study, conducted by data analysis experts, demonstrates that Decision Company is capable of producing diverse and high-quality data, laying the groundwork for efficient data annotation. We evaluate popular and advanced LLMs in Tapilot-Crossing, which highlights the challenges of conversational tabular data analysis. Furthermore, we propose **A**daptive **C**onversation **R**eflection (**ACR**), a self-generated reflection strategy that guides LLMs to *learn from successful histories*. Experiments demonstrate that ACR can evolve LLMs into effective conversational data analysis agents, achieving a relative performance improvement of up to 44.5%.

## 1 Introduction

The exponential growth of big data calls for accessible data analysis techniques that cater to a wide range of applications, such as healthcare, games, and entertainment (Khanbabaei et al., 2018; Han et al., 2011; Fayyad et al., 1996). Recently, the development of LLM agents (Liu et al., 2023c; Xu et al., 2023b; Zeng et al., 2023; Xu et al., 2023a; Deng et al., 2024; Si et al., 2023) has attracted a lot of attention. They are capable of understanding natural language queries, as well as generating codes for data manipulation and visualization, through reasoning (Huang & Chang, 2023; Wei et al., 2022; Yao et al., 2023) and tool calls (Li et al., 2023b; Huang et al., 2023c; Qin et al., 2023). Among the vast types of data available, tabular data stands out as one of the most prevalent and interpretable formats organized by rows and columns.

Tabular data analysis agents, such as SheetCopilot (Li et al., 2024c), TableGPT (Zha et al., 2023), and Data-Copilot (Zhang et al., 2023a), provide automatic workflow based on user queries. However, the dynamic and uncertain nature of real-world analysis hinders effective human-agent conversation, since user intents can often be ambiguous (De Vries et al., 2020; Yan et al., 2023; Wang et al., 2024), and users may need to adjust their analysis strategies based on intermediate results (Yan et al., 2023; Yao et al., 2020). For example, in Figure 1, the `notable opponents` could refer to a variety of interpretations, such as the opponents with the highest wins, or the most frequent opponents. Towards this end, a comprehensive benchmark is indispensable for gauging their capability in conversational user engagement within data analysis scenarios.

In this paper, we introduce **TA**ble co**PILOT Crossing** (Tapilot-Crossing), a new benchmark for evaluating LLM agents in conversational data analysis tasks. Tapilot-Crossing is designed to simulate real-world data analysis scenarios, where users converse with LLM agents to generate codes for data exploration and decision making. It includes 1024 user-machine conversations with 1176 user intents, spanning four practical scenarios, as shown in Figure 1: **1) Normal** mode refers to

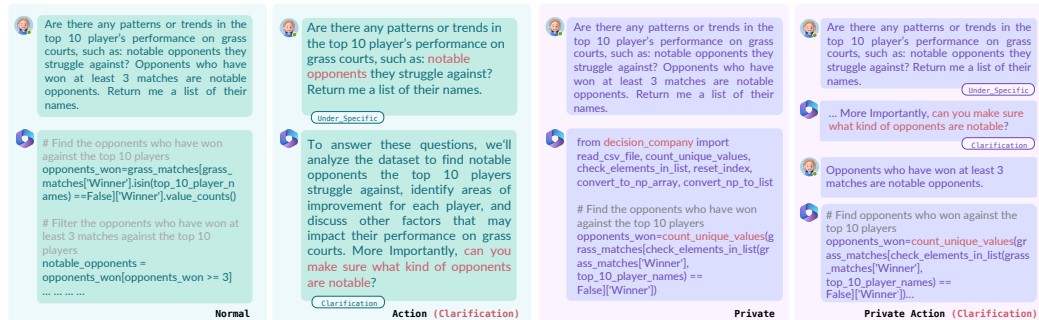

Figure 1: An overview of the four conversation modes in TAPILOT-CROSSING, illustrated by relevant aspects of the associated codes or actions. The term `Notable Opponents` exemplifies an ambiguous term that necessitates clarification through multi-turn conversations. `count_unique_values()` is one example function from unseen private libraries.

the scenario where all questions and user requirements are explicit, agents can answer questions by referring only to table contents and dialog histories. This would evaluate fundamental capabilities of agents in handling data analysis tasks; **2) Action** mode represents that agents must infer diverse user intents first to deliver satisfactory results. For example, they need to interpret ambiguous terms such as `notable opponents` by asking questions and generate appropriate responses based on user clarification. This tests their ability to respond to complex and dynamic user queries during conversations; **3) Private** mode is designed to examine the true semantic parsing capability of agents when encountering unseen packages provided by users (Zan et al., 2022); and **4) Private Action** mode unifies the challenges of Private and Action modes, more closely reflecting real-world data analysis. Answer types can be summarized into two categories: **1) Code Generation**, which can test whether the agent can correctly interpret the user query and generate the corresponding code for data analysis, and **2) Multiple-Choice questions**, which can evaluate the ability of agents to understand the returned results being executed codes and provide users with appropriate insights.

The conventional construction of datasets or benchmarks based on crowdsourcing, particularly for high-quality and conversational scenarios, is time-consuming and costly due to the significant human effort and expertise required (Yu et al., 2019a; Li et al., 2023a; Guo et al., 2021; Li et al., 2023d; Zhang et al., 2023d). In this case, we design a novel multi-agent environment, called **DECISION COMPANY**, to construct TAPILOT-CROSSING. DECISION COMPANY is a simulated environment where four GPT-4 agents communicate with each other to perform data analysis tasks. Utilizing this environment allows us to construct TAPILOT-CROSSING within a month at a cost of less than $100. We also design an evaluation script caching method to bind concurrent human-crafted high-quality evaluation scripts to new data, in order to enable the scalability of TAPILOT-CROSSING. The reliability and potential biases of such human-AI collaborative approach are rigorously evaluated through human evaluation involving 10 data analysis experts in terms of general and action-wise dataset study. The result shows compelling evidence that DECISION COMPANY cannot only scale effectively but also maintain exceptional data quality.

We evaluate the popular advanced LLM agents on TAPILOT-CROSSING. The results underscore the challenges of conversational data analysis and fuel the need for more advanced LLM agents that can handle diverse user intents and feedback. To further evolve the LLMs towards effective conversational data analysis agents, we propose **A**daptive **C**onversation **R**eflection (**ACR**), which guides LLM agents to *learn from successful history* via self-generated pseudo logic reflection. Our experiments demonstrate that **ACR** can significantly enhance the performance of LLMs, in which GPT-4 can gain relative improvement of **44.5%** compared to its model base, offering an insight of how to actively improve the conversation between human and LLM agents in data analysis tasks.

## 2 PRELIMINARIES

**Task Formulation.** Conversational data analysis with LLM agents involves a sequence of user-agent turns, $[(u_1, a_1), (u_2, a_2), \ldots, (u_n, a_n)]$, where each turn $(u, a)$ consists of a user query $u$ and an agent response $a$. Queries can be instructions or feedback, while responses can be code snippets

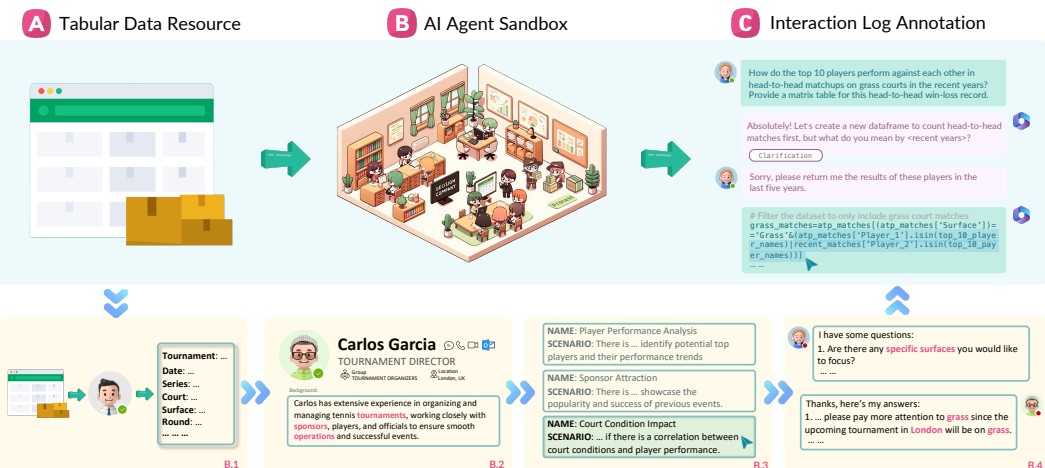

Figure 2: The construction pipeline of TAPILOT-CROSSING by the AI Agent Sandbox DECISION COMPANY. ◥ denotes human intervention during construction.

or selected answers. Dialogues start with $u_1$ and end with $a_n$. Given the current user query $u_t$, all previous user-agent history $H$ from turn 1 to $t-1$, and sampled table contents $T$, the agent should act, such as asking for clarification, and generate an answer $a_t = f_\theta(u_t, H, T)$, where $f_\theta$ refers to the agent built based on LLMs with model weights $\theta$. This setup allows TAPILOT-CROSSING to evaluate conversational agent performance in a static and systematic manner. Please note that during evaluation in ACTION mode, each action is tested individually to evaluate how well the model performs for each action type. A more dynamic setting, where models must first select their own actions then perform analysis tasks, will be explored in future work.

**Action Types.** We identify 6 common agent actions during conversations, each serving as a specific evaluation mode in TAPILOT-CROSSING. The agent actions in TAPILOT-CROSSING include Update_Code, which addresses user requests for bug fixes or refinements; Fast_Fail, which alerts users to insufficient data or factual errors; and Clarification, where agents seek additional information for under-specified queries. To reduce user impatience and long dialogue issues, Best_Guess allows agents to make assumptions based on data, domain knowledge, and common-sense, though it risks incorrect guesses. The Plot_QA action helps users understand plot-derived insights, while Insight_Mining involves summarizing executed results to aid in decision-making, evolving agents into comprehensive data analysis tools. Detailed examples and interpretations are in Appendix K, and evaluation methods for each mode are provided in Appendix M.

## 3 THE TAPILOT-CROSSING DATASET

### 3.1 DATASET CONSTRUCTION OVERVIEW

The construction of TAPILOT-CROSSING is mainly based on the AI Agent Sandbox, DECISION COMPANY, as depicted in Figure 2. DECISION COMPANY is a multi-agent environment where four GPT-4 agents (Administrator, Client, Data Scientist and AI Chatbot) converse with each other to perform data analysis tasks. The construction process involves the following steps: Data Acquisition & Preprocessing, Client Persona Generation, Analysis Scenario Generation, Plan Discussion and conversation Log Simulation. During these stages, human intervention may be required to correct errors or eliminate harmful or biased content.

**Data Acquisition & Preprocessing.** The first step in the construction of TAPILOT-CROSSING is the acquisition and preprocessing of data. We collect open-source tables from Kaggle [1], a popular data science platform. These tables cover 18 analysis topics under 5 common domains, namely ATP Tennis, Credit Card, Fast Food, Laptop Price, and Melbourne Housing,

---

[1]https://www.kaggle.com/

as detailed in Appendix C.3. Given any of the tables, Administrator Agent will generate column meanings and value illustrations.

**Client Persona Generation.** The construction of TAPILOT-CROSSING proceeds to the generation of client personas. These personas with specific tasks and topics related to the data are created by the Administrator Agent. Each persona is defined by a `Name`, `Location`, `Job`, and `Background` with a diverse range of interests and backgrounds.

**Simulation of Analysis Scenarios.** Then, Administrator Agent conducts interviews with each Client Agent to ask about their `Scenario` description, `Scenario Name`, and the `Goal` of using the dataset for the scenario. In the TAPILOT-CROSSING dataset, human annotators will intervene here and select the most reasonable or interesting scenarios. This ensures that the scenarios included in the TAPILOT-CROSSING dataset are meaningful, and not too general across different clients. For instance, in B.3 of Figure 2, we select `Court Condition Impact` because `Player Performance Analysis` is too general and `Sponsor Attraction` requires too much additional information out of the table contents, which leads to too many unanswerable questions.

**Plan Discussion.** In this process, the Data Scientist Agent engages with the Client Agent to convert the requirements of client into a series of specific data analysis questions with well-defined conditions. Each question is provided by an expected result type, such as dataframes, lists, or various plot types, which helps reduce question ambiguity and ease the pressure on evaluation metrics (Yin et al., 2023; He et al., 2023; Zhang et al., 2023c). The dialogue between the agents further refines the questions with specific conditions. For example, as depicted in Figure 2 B.4, the client `Garcia`'s question could be further elaborated on the basis of his following responses, making all questions more answerable. In particular, Agent `Garcia`, fully cognizant of his persona created in B.2, adds the condition `grass`, reflecting his `London` location. This implies that the role-playing aspect of the agent can be instrumental in generating a wider range of questions that are both diverse and reasonable (Li et al., 2024a; Park et al., 2023).

**Conversation Log Annotation.** Following the plan discussion, the conversation simulation phase begins. Here, the AI Chatbot Agent takes the lead, executing the data analysis plan agreed on during the previous stage. The AI Chatbot Agent converses with the Data Scientist Agent to answer a series of questions defined in the plan by generating codes and analyzing returned results.

## 3.2 HUMAN CALIBRATION AND ANNOTATION

While the DECISION COMPANY can generate a wealth of data analysis conversations in a zero-shot prompting manner, human intervention is indispensable to ensure the quality of the data set annotation (Lu et al., 2023). Our observations indicate that only 23.5% of the original responses produced by the AI Chatbot Agent can be directly used as reference codes. Therefore, we engage two PhD students in each stage of the generation process to calibrate the errors and meaningless conversations. While human intervention is required, it is worth notice that modifying existing answers or codes is more efficient than creating them from scratch. We preserve all natural and meaningful conversations, both agent-to-agent and human-to-machine, throughout the action setting collection.

The involvement of two PhD students follows a careful workflow to ensure the quality and accuracy of the benchmark data. Each student, with over 10 years of experience in data analysis, is assigned with distinct sets of initial data analysis scripts consisting of topics, tabular data, and first-turn questions. During each conversational turn, the AI Chatbot generates code in response to specific analysis questions. The students then execute this code, verify the outputs for correctness, fix any bugs, and ensure that the results align with the expected output types, which can answer the question accurately. This refined code is referenced as ground-truth code for the benchmark. Additionally, the students generate evaluation scripts (`eval.py`) for testing model-generated code. Following this, the students execute the code and feed the results back to the AI Chatbot, which generates free-form text analysis statements (e.g., `"The credit card application rate of people with 4-year employment is 73.5% higher than those with no employment"`). The students review and correct these statements, converting them into multiple-choice question formats for more reliable and objective evaluation. All corrected code and free-form text analysis are stored as static user-AI conversation history, enabling further reference during annotation. After completing this conversational process, the students exchange their annotated data for cross-validation, achieving

| Dataset | # Q \| # Intents | # Toks. / Q | # Toks. / Code | Code Type | Analysis | Multi-Turn | Private Lib | Multi-modal | Evaluation |
|---|---|---|---|---|---|---|---|---|---|
| HumanEval (Chen et al., 2021) | 164 \| 164 | 60.9 | 24.4 | 🐍 | ✗ | ✗ | ✗ | ✗ | Test Cases |
| MBPP (Austin et al., 2021) | 974 \| 974 | 14.5 | 24.2 | 🐍 | ✗ | ✗ | ✗ | ✗ | Test Cases |
| Spider (Yu et al., 2018) | 1034 \| 1034 | 12.4 | 18.3 | 🗄 | ✗ | ✗ | ✗ | ✗ | Acc + EM |
| BIRD (Li et al., 2023a) | **1534 \| 1534** | 14.5 | 49.6 | 🗄 | ✗ | ✗ | ✗ | ✗ | Acc + VES |
| DS-1000 (Lai et al., 2023) | 1000 \| 1000 | **282.4** | 42.1 | 🐍 | ✗ | ✗ | ✗ | ✓ | Test Cases + SFC |
| SparC (Yu et al., 2019b) | 1203 \| 1203 | 9.4 | 26.3 | 🗄 | ✗ | ✓ | ✗ | ✗ | Acc |
| CoSQL (Yu et al., 2019a) | 1008 \| 1008 | 13.1 | 31.4 | 🗄 | ✗ | ✓ | ✗ | ✗ | Acc |
| ARCADE (Yin et al., 2023) | 1066 \| 1066 | 19.2 | 48.2 | 🐍 | ✗ | ✓ | ✗ | ✗ | Acc + Fuzzy |
| **TAPILOT-CROSSING** | 1024 \| 1176 | 273.6 | **110.6** | 🐍 | ✓ | ✓ | ✓ | ✓ | Acc + AccR |

Table 1: Comparison of TAPILOT-CROSSING and other data analysis datasets. The first 5 datasets are single-turn data analysis sets featuring both SQL and Python codes. The following 3 benchmarks are multi-turn or conversational data analysis datasets. TAPILOT-CROSSING represents a challenging dataset in data analysis with more comprehensive settings. 🐍 represents that the end code is Python. 🗄 means the target code is SQL.

an inter-agreement score of 93.64%. This rigorous process is crucial, especially given the more objective and quantitative nature of data analysis tasks compared to standard NLP tasks.

### 3.3 PRIVATE LIBRARY MODE EVOLUTION

Data analysts frequently rely on their private libraries (Zan et al., 2022). These libraries, often tailored for specific needs, allow more efficient and customized data processing and analysis. Furthermore, generating codes by user-defined packages can test the true semantic parsing abilities of agents rather than merely testing their memorization of standard syntax from libraries such as Pandas or Numpy. It also evaluates their ability to understand and implement custom functions, which is a crucial aspect of real-world data analysis. In this work, we prompt GPT-4 to autonomously convert prototype codes with pre-trained functions like Pandas and Numpy into private codes. It contains three steps: 1) summarizing packages from original code; 2) refactoring and converting them to user customized functions with light human supervision; 3) regenerating codes via customized functions with human calibration. Please refer to Appendix T for more details.

### 3.4 EVALUATION SCRIPT CACHING

Each example will be provided by a specific evaluation script to ensure the precision of our assessments (Lai et al., 2023). To manage the extensive effort required to design scripts for each example, we introduce a cache-based evaluation binding approach. Initially, we classify mainstream result types, which are collected through steps introduced in Section 3.1, and develop highly specialized scripts for each type, such as dataframes and dictionaries. When new data is generated, an evaluation script is automatically assigned based on the result type. Two PhD students then review the assigned script to ensure its accuracy; if necessary, they adapt and generate a new script tailored to the specific case. This method allows us to streamline the evaluation process, making it more efficient with minimal human intervention. The details of the evaluation script in each result type can be found in Section L.

## 4 DATA STATISTICS & METRICS

### 4.1 DATASET STATISTICS

Figure 3 provides key statistics for our dataset, while Table 1 offers a comparison between TAPILOT-CROSSING and other datasets related to data analysis. To ensure a fair comparison regarding question and code length, we utilize tiktoken[2] to compute the number of tokens for each dataset. As shown in Table 1, TAPILOT-CROSSING includes comprehensive evaluation settings across private library, multi-turn, and multi-modal conversations. Besides, the complexity of this dataset, reflected by the long questions and their associated code snippets, is amplified by the inclusion of multi-intent queries. These queries, encapsulating multiple intents within a single

---

[2] https://github.com/openai/tiktoken

Figure 3: Data characteristics

| Statistic | Number |
|---|---|
| **Total conversations** | 1024 |
| ⟩ clear conversations | 284 |
| ⟩ action conversations | 485 |
| ⟩ private lib. conversations | 206 |
| ⟩ private act. conversations | 49 |
| ⟩ # of private lib functions | 137 |
| **Answer Types** | |
| ⟩ # of code generation answers | 594 |
| ⟩ # of multi-choice answers | 430 |
| **Quality & Cost** | |
| ⟩ inter-agreement | 93.64 |
| ⟩ total costs (USD) | 66.7 |
| ⟩ AVG # of turns | 14.15 |

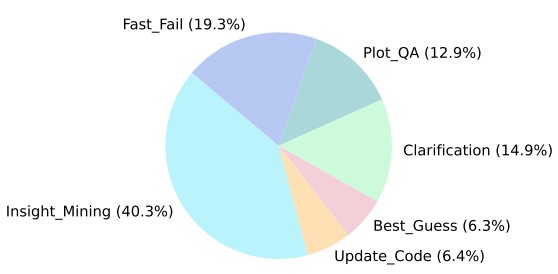

Figure 4: Distribution of ACTION Mode in TAPILOT-CROSSING. It contains 6 common types in conversational data analysis tasks.

question, require a versatile array of computational strategies for effective handling. For example, the query, `"Please provide histogram plots and mean for employment status of credit card applicants."` demands both data visualization and statistical evaluation. Finally, although TAPILOT-CROSSING contains 1,024 data analysis conversations, it only incurs a cost of 66.7 USD, making it an economical choice for dataset generation. The inter-agreement of 93.64 promises the high quality of the dataset.

| Human Calibration | Conversation | | | | | Eval Scripts | |
|---|---|---|---|---|---|---|---|
| | Conversation Coherence | Scenarios Diversity and Reasonableness | Conversation Topic Coherence | Ethics and Bias Representation | Conversation Naturalness | Evaluation Scripts Quality | Evaluation Scripts Scalability |
| Before | 0.19 | 0.46 | 0.17 | 0.41 | 0.67 | - | - |
| After | **0.97** | **0.96** | **0.93** | **1.00** | **0.95** | **0.98** | **0.94** |

Table 2: Acceptance ratio of human evaluation on general metrics of the dataset quality, **the higher the better**. The table reports the percentage of samples considered qualified or being accepted for each metric.

## 4.2 EVALUATION METRICS

**Accuracy (Acc).** Acc is a metric that evaluates the ability of agents in generating codes that execute correctly or answer multiple-choice questions accurately. It is defined as the proportion of instances whose predicted outputs match the expected reference output, across all evaluated tasks.

**Accuracy with Private Library Recall (AccR).** Recognizing the importance of accurately leveraging specific user-defined libraries in code generation, we extend Acc to include a recall-based adjustment for instances involving private libraries. This ensures that AccR not only evaluates the direct accuracy of code execution and question answering but also the inclusion and correct usage of private library functions. We conduct an in-depth analysis of the impact of AccR on Private mode evaluation in Section N.

## 5 DATASET QUALITY EVALUATION

To ensure the data quality of TAPILOT-CROSSING and the reliability of our proposed data generation workflow named DECISION COMPANY, we conduct a comprehensive human evaluation focusing on both general and action-specific aspects by selecting 500 samples which is approximately 50% of the full dataset.

**General Metrics.** Following (Hu et al., 2024), we conduct human evaluation on more NL metrics about reasonableness and coherence across turns of conversations. The results are shown in Table 2, where `Before` refers to the pre-calibration data is fully annotated by LLM agents. And `After` means the data in the TAPILOT-CROSSING, which is annotated by human-AI collaboration. In the table, there is an obvious improvement in dataset quality following human calibration performed by two PhD students. The low acceptance ratio of the data prior to calibration underscores the necessity of this process. After calibration, the acceptance ratio rises to approximately 0.95, indicating that the

involvement of two PhD students who are professionals in data analysis is sufficient to ensure the quality of the dataset, thus demonstrating the balance of trade-off in DECSION COMPANY between efficiency and quality of complex data annotation workflow. All details can be found in Appendix Q.

**Action-wise Metrics.** Human evaluation is also conducted with a focus on the actions. Figure 5 illustrates the consensus among experts that all actions in TAPILOT-CROSSING are both necessary and commonly observed in real-world data analysis scenarios. All instructions and details can be found in Appendix Q.

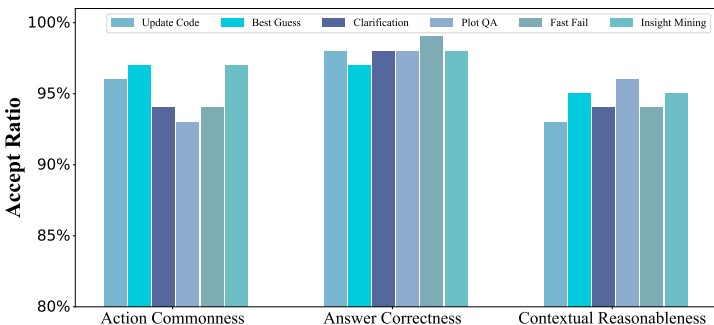

Figure 5: Results of human evaluation on action-wise metrics of the dataset quality.

# 6 EVOLVING LLMS TOWARDS CONVERSATIONAL DATA ANALYSIS AGENTS

In this section, we discuss our approach of equipping LLMs as data analysis agents with tools and reasoning. We then introduce our self-generated reflection strategy, ACR, to enhance their performance in conversational settings.

## 6.1 TOOLKIT

Our tool sets include an executor, a user simulator, and a chart-to-table converter. The executor provides an environment for models to observe real-time feedback on their intermediate code results (Xie et al., 2023; Wang et al., 2024). The user simulator (Wang et al., 2024; Yan et al., 2023), powered by GPT-4-Turbo, tests the agents' ability to generate codes after clarifying details when facing under-specific questions. The chart-to-table (Liu et al., 2023a) converter mitigates the prevalent issue of LLMs' inability to comprehend plots by converting them into tables. Detailed descriptions of these tool sets can be found in Appendix I.2.

## 6.2 REASONING

Reasoning is a critical process for transitioning LLMs into data analysis agents (Huang & Chang, 2023). In TAPILOT-CROSSING, we incorporate two primary reasoning methods for code generation and multiple-choice answers. The first is the **Chain-of-Thought (COT)** prompting technique (Wei et al., 2022), which enhances the complex reasoning abilities of LLMs by dividing the reasoning path into multiple steps. The second method is **Reasoning & Action (ReAct)**, which enables models to make decisions by generating reasoning traces and actions in an interleaved manner, inducing writing, executing them, understanding results, and analyzing them to make the final choices (Yao et al., 2023).

## 6.3 ADAPTIVE CONVERSATION REFLECTION (ACR)

Successful conversations are important since they encapsulate the logic necessary to meet user requirements and ensure correct steps of analysis or code generation. Motivated by this, we propose the **A**daptive **C**onversation **R**eflection (**ACR**) approach to enable data analysis agents to learn from successful user-code histories through a two-step process.

| Model | Conversation Mode | | | | Answer Type | | Overall |
|---|---|---|---|---|---|---|---|
| | Normal | Action | Private | Pri-Act | Code | Choice | |
| Mistral-7B | 5.4 | 15.8 | 1.0 | 1.3 | 3.5 | 15.8 | 8.7 |
| Claude-2.1 | 20.2 | 16.8 | 1.5 | 4.5 | 11.4 | 17.9 | 14.1 |
| Mistral-8 × 7B | 16.0 | 22.6 | 2.9 | 2.2 | 9.8 | 24.9 | 16.1 |
| CodeLlama-34B | 27.5 | 18.7 | 2.4 | 0.0 | 15.0 | 19.8 | 17.0 |
| GPT-4-Turbo | 27.6 | 17.5 | 5.3 | 4.3 | 17.9 | 16.1 | 17.1 |
| Llama-3-8B | 28.8 | 24.4 | 1.1 | 2.9 | 16.0 | 25.5 | 20.0 |
| Claude-3-Opus | 21.5 | 28.2 | 2.2 | 5.4 | 13.3 | 29.5 | 20.1 |
| Llama-3-70B | 30.8 | 35.0 | 2.8 | 0.0 | 17.7 | 37.0 | 25.8 |
| GPT-4-32k | 29.7 | 24.2 | 7.1 | 0.0 | 17.8 | 25.4 | 21.0 |
| + Agent | 23.4 | 39.2 | 9.1 | 5.3 | 16.6 | 38.8 | 25.9 |
| + Inter-Agent | **32.2** | **41.3** | **10.6** | **9.8** | **21.6** | **42.1** | **30.2** |

Table 3: Overall results of LLMs in base, agent, and inter-agent modes on the TAPILOT-CROSSING dataset. **Pri-Act** refers to private library + action evaluation mode.

**Pseudo Code Logic Generation.** First, given the last previous history $(\mathbf{u}_{t-1}; \mathbf{a}_{t-1})$, when $t > 1$, we prompt the data analysis agent to reflect and generate its underlying logic $\mathbf{m}_{t-1} = f_\theta(\mathbf{u}_{t-1}; \mathbf{a}_{t-1})$, where $f_\theta$ refers to agent based on LLMs with parameter $\theta$. Also, $(x; y)$ represents two elements $x$ and $y$ are concatenated in the prompt. In our work, we consider the pseudocode to be $\mathbf{m}$, as it serves as an intermediate logic between natural language queries and codes.

**Re-Org One-Shot Reasoning.** Second, we re-organize them into a self-generated one-shot example with the order: $\mathbf{p}_{t-1} = (\mathbf{u}_{t-1}; \{\mathbf{m}_{t-1}; \mathbf{a}_{t-1}\})$, which represents the scenario where the input $\mathbf{u}_{t-1}$ is given, the agent should generate a logic $\mathbf{m}_{t-1}$ first, then generate answers $\mathbf{a}_{t-1}$. Finally, the data analysis agent can learn from $\mathbf{p}_{t-1}$ to first generate logic $\mathbf{m}_t = f_\theta(\mathbf{p}_{t-1}; \mathbf{u}_t)$ and generate an answer $\mathbf{a}_t = f_\theta(\mathbf{u}_t; \mathbf{m}_t)$ in the current turn $t$. When $t = 1$, we keep the same reasoning method of the original agent. Appendix H provides a detailed example for further illustration.

# 7 EXPERIMENTS

## 7.1 SETUP

**Models.** Our experiments primarily involve popular LLMs that are capable of generating code and following complex human instructions since this is a basic requirement in data analysis. Therefore, we investigate performance of 4 families of models, covering Mistral (Jiang et al., 2023), LLama (Roziere et al., 2023; Dubey et al., 2024), Claude [3], and GPT (Achiam et al., 2023) models. Appendix E contains details of the model alias.

**Implementation Details.** The implementations could be divided into three settings: 1) `Model-Base` refers to the LLM itself without reasoning and tool calls. 2) `Agent` mode involves multiple tool usage and reasoning. We employ zero-shot COT for guiding the LLM in code generation tasks, as it allows us to test the agents' pure code generation abilities in data analysis. For multi-choice question answering, we utilize one-shot ReAct. 3) `Inter-Agent` mode incorporates ACR as described in Section 6.3 beyond the AGENT. Each model is provided with the last up to 5 turns of user-code histories. Further details can be found in Appendix J. For the implementation of PRIVATE settings, we follow (Li et al., 2023b; Zan et al., 2022) to prompt agents to retrieve private libraries first then generate the code with retrieved packages. Due to the budget constraints, we implement `Model-Base` modes for all LLMs and `Agent` and `Inter-Agent` models for Mistral, LLama, Claude, and GPT model families.

## 7.2 EXPERIMENTAL RESULTS

**Overall Results.** Table 3 illustrates the comprehensive performance of many LLMs and two developed agent modes based on GPT-4-32k on the TAPILOT-CROSSING dataset. From the results, we can deduce the following: 1) TAPILOT-CROSSING is a challenging benchmark in which the SOTA method GPT-4-32k w/ `Inter-Agent` only achieves a score of 30.2, leaving a large room for

---
[3]https://claude.ai/

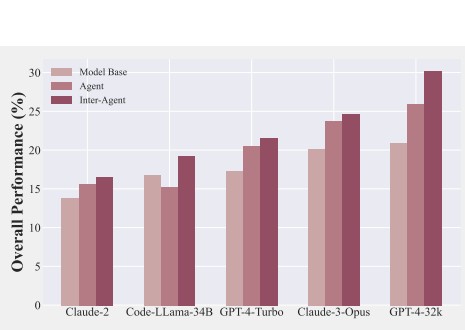

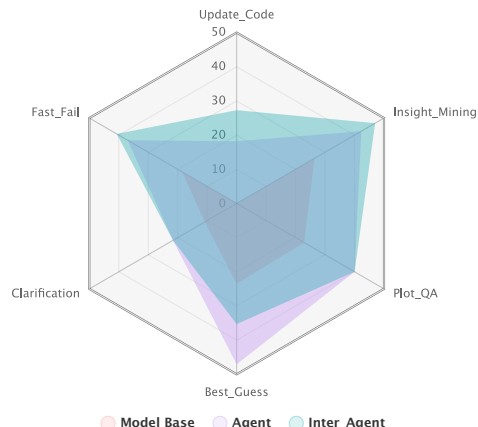

(a) Visualization of the performance of Claude-2, Code-Llama-34B, GPT-4-Turbo, Claude-3-opus, and GPT-4-32k, all with base, agent, and inter_agent versions. For each model, the left bar is the base version, the middle bar is the agent version, and the last bar is the inter_agent version.

(b) Visualization of the performance of GPT-4-32k across various categories in ACTION Mode. The comparison includes base, agent, and inter_agent versions.

Figure 6: Visualization of the performance of LLMs with base, agent, and inter_agent versions.

improvement. 2) Despite the performance of GPT-4-Turbo being nearly on par with GPT-4 in code generation, its overall performance still falls short of GPT-4. This indicates that beyond code writing, understanding results, and analysis are equally important. Fortunately, the comprehensive settings of TAPILOT-CROSSING can assist users in selecting models for data analysis tasks. 3) It is surprising to see the exceptional performance of open-sourced models, such as CodeLlama and Llama-3-70B, in the NORMAL code generation setting. We observe that CodeLlama frequently defines functions automatically and applies these in the following code, thereby improving readability and logic. This is particularly beneficial in tasks related to data-analysis code generation. Such tasks often require the composition of API functions, which demands a profound understanding of the context and the ability to extract common patterns into reusable functions. By defining and reusing symbolic functions, CodeLlama can streamline complex contexts, making them more logical, which is an advantage for resolving complex tasks (Gu et al., 2023). 4) Llama-3-70B performs better than GPT-4 on base mode proving that our benchmark is not overfitting to the GPT family of models.

**LLM Agent Performance.** We also implement different agent modes for several mainstream LLMs. As shown in Figure 6(a), most models with Agent version outperform their base version, highlighting the crucial role of tools and reasoning in enhancing the performance of LLMs under complex tasks (Liu et al., 2023c; Xie et al., 2023). Also, all models exhibit obvious improvements in the Inter-Agent mode with ACR. This indicates that the underlying logic of successful conversation histories is instrumental in guiding LLMs to become more proficient data analysis agents in conversational settings.

**Fine-Grained Results on ACTION Modes.** Figure 6(b) provides a comparative evaluation of three GPT-4 model variants across various ACTION modes detailed in Section 2. The conversational data analysis agent, Inter-Agent, obviously outperforms in most areas, especially in managing Fast_Fail queries and executing Update_Code actions. However, it falls short in the Best_Guess action when compared to the Agent. We note that ACR tends to make agents overly tractable in re-org one-shot example $\mathbf{p}_{t-1}$ and current generated logics $\mathbf{m}_t$. If $\mathbf{p}_{t-1}$ and $\mathbf{m}_t$ do not contain instructions on making assumptions, agents tend to select None of Above. This observation suggests that excessive reliance on historical data may hinder the inherent ability of models to conjecture based on instant user behaviors. Therefore, striking a trade-off between user-code history exploration and real-time user conversation, especially when facing under-specific questions, is crucial for improving the performance of LLM agents in conversational settings.

**Error Analysis.** We conducted an error analysis by sampling 200 error cases from each of 5 LLMs to gain insights into conversational data analysis. A detailed analysis is available in Appendix

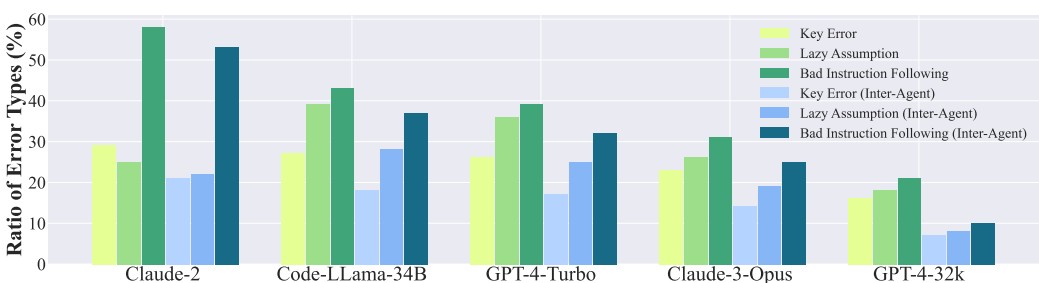

Figure 7: The visualization of different error types across different models and settings.

O. The errors were categorized into three main types: **(1) Key Error (21%)**, where the model incorrectly matches column names in the provided data table or references nonexistent values; **(2) Lazy Assumption (37%)**, where the model assumes that intermediate results or states are already available or saved on disk without verification; and **(3) Poor Instruction Following (56%)**, where the model fails to strictly adhere to the given instructions, resulting in incorrect or incomplete answers. Furthermore, Figure 7 demonstrates that our ACR method effectively reduces errors in each category.

## 8 RELATED WORK

The use of LLMs for data analysis has garnered significant interest, with In-Context Learning applied to tasks like SQL query generation (Pourreza & Rafiei, 2024a; Gao et al., 2023; Lei et al., 2023; Zhang et al., 2023b; Gu et al., 2024; Wang et al., 2023a; Pourreza & Rafiei, 2024b; Li et al., 2024b), pandas or Python code generation (Jain et al., 2023; Chen et al., 2024; 2023a; Li et al., 2024c; Zha et al., 2023; Zhang et al., 2023a; Zheng et al., 2024b), and data visualization (Chen et al., 2023b; Huang et al., 2023a). While most studies focus on single-turn conversations with explicit queries, recent work highlights the need for conversational data analysis to refine user intents (De Vries et al., 2020; Yan et al., 2023; Wang et al., 2024). Benchmarks like HumanEval (Chen et al., 2021) and Spider (Yu et al., 2018) for single-turn, and CoSQL (Yu et al., 2019a) and ARCADE (Yin et al., 2023) for multi-turn conversations, primarily target code generation, while often neglecting data visualization and intermediate result understanding. Our benchmark, TAPILOT-CROSSING, addresses this gap by evaluating LLM agents in conversational data analysis using the multi-agent environment DECISION COMPANY. Inspired by multi-agent environments for data generation (Lu et al., 2023; Ding et al., 2023; Li et al., 2023b; Park et al., 2023), we generate realistic conversation logs for data analysis, thus pioneering an conversational benchmark for evaluating data analysis agents.

## 9 CONCLUSION

We introduce TAPILOT-CROSSING, a new benchmark for evaluating LLM agents in conversational data analysis tasks. TAPILOT-CROSSING is constructed via a cost-effective and expert-recognized high-quality multi-agent environment, DECISION COMPANY, and covers a wide range of practical scenarios. We evaluate data analysis agents based on popular LLMs on TAPILOT-CROSSING, highlighting the challenges of conversational data analysis and the need for more advanced conversational data analysis agents. We also propose ACR, an effective reflection strategy for conversational data analysis agent evolution. Our experiments demonstrate that ACR can significantly enhance the performance of LLM agents.

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

## A WHY NAMED TAPILOT-CROSSING?

TAPILOT-CROSSING means benchmarking table copilot for complex data analysis tasks by conversing with humans. It is inspired by the popular Switch game, Animal-Crossing. where users can perform complex tasks such as constructing fantastic architectures through conversations with animal citizens (agents).

## B BACKGROUND KNOWLEDGE

**Requirements of client.**   It naturally refers to specific data analysis tasks or questions that users want to accomplish, expressed in natural language.

**Result Type.**   It typically refers to the format or nature of the output produced from analyzing data. Common result types include: "dataframes, lists, or various plot types."

**User intent.**   It represents the underlying analytical goal or purpose behind a user's query.

## C DATA RESOURCE

### C.1 TAPILOT-CROSSING

Our TAPILOT-CROSSING data is available under the lisense CC BY-SA 4.0.[4]

### C.2 KAGGLE TABULAR DATA

The tabular data that we used to create TAPILOT-CROSSING are following open-source licenses: 1) **Public Domain**: Public Domain Mark 2) **CC-BY**: Creative Commons Attribution 4.0 International.

### C.3 DATA DISTRIBUTION

The Figure 8(a) visualizes our covered topics and domains.

## D COMPARISON WITH AGENT-RELATED BENCHMARKS

Besides the benchmark comparisons in Table 1, we list more popular LLM Agent-related benchmarks comparisons in Table 4. Each column means: (1) **#Q**: This column represents the unique identifier or number assigned to each benchmark or dataset. (2) **eval public available**: This column specifies whether the evaluation metrics of the benchmark or dataset is publicly available for use. (3) **multi-modal**: This column shows whether the benchmark supports multi-modal data, meaning it can handle multiple types of input data (e.g., text, images) simultaneously. (4) **private lib**: This column indicates whether the benchmark or dataset includes a private library. (5) **Multi-Turn**: This column specifies whether the benchmark supports multi-turn conversations, which are conversations that involve multiple exchanges or steps. (6) **conversation trajectory**: This column indicates whether the benchmark involves conversation trajectories, which track the sequence and flow of conversations over time. (7) **data creation**: This column describes the method of data creation for the benchmark. "From-scratch" means the data was created anew specifically for the benchmark, while "semi" indicates that the data was created using a mix of new and existing data. (8) **output type**: This column specifies the type of output produced by the benchmark or dataset. Examples include "Text," "Multi-Types," "Patch," "Code," and "Code/Choice," indicating the nature of the outputs generated during evaluations.

---

[4]https://creativecommons.org/licenses/by-sa/4.0/deed.en

| | # Q | eval public available | multi-modal | private lib | multi-turn | conversation trajectory | data creation | output type |
|---|---|---|---|---|---|---|---|---|
| GAIA (Mialon et al., 2023) | 466 | ✗ | ✓ | ✗ | ✗ | ✗ | from-scratch | Text |
| ResearchAgent (Huang et al., 2023b) | 15 | ✓ | ✓ | ✗ | ✗ | ✗ | semi | Multi-Types |
| SWE-bench (Jimenez et al., 2023) | 2290 | ✓ | ✗ | ✗ | ✗ | ✗ | from-scratch | Patch |
| AgentBench (Liu et al., 2023c) | 1091 | ✓ | ✓ | ✗ | ✓ | ✗ | semi | Multi-Types |
| RepoBench (Liu et al., 2023b) | 1669 | ✓ | ✗ | ✗ | ✗ | ✗ | semi | Code |
| DebugBench (Tian et al., 2024) | 4253 | ✓ | ✗ | ✗ | ✗ | ✗ | from-scratch | Code |
| **TAPILOT-CROSSING** | 1024 | ✓ | ✓ | ✓ | ✓ | ✓ | from-scratch | Code/Choice |

Table 4: Comparison of TAPILOT-CROSSING to popular LLM Agent-related datasets. TAPILOT-CROSSING represents a challenging dataset in data analysis with more comprehensive settings.

# E   MODEL DESCRIPTIONS

In this section, we provide an overview of the various models used in our research. These models include both widely recognized and state-of-the-art LLMs that have been instrumental in advancing NLP tasks.

(1) **Mistral-7B-instruct-v01**: Mistral-7B is a powerful LLM designed to handle diverse NLP tasks. It is known for its efficiency in terms of parameter size while maintaining high performance. The 7 billion parameters enable it to process and generate human-like text effectively.

(2) **Claude-2.1**: Claude-2.1 is an advanced version of the Claude series of LLMs. This iteration brings improvements in both accuracy and processing speed, making it a suitable choice for complex language understanding and generation tasks.

(3) **Mistral-8 × 7B-instruct-v01**: Mistral-8 × 7B represents a collection of 8 models, each with 7 billion parameters. This ensemble approach allows for enhanced performance through model averaging and provides robustness in generating more accurate results across different tasks.

(4) **CodeLlama-34B-Instruct-hf**: CodeLlama-34B is a specialized model focused on code-related tasks. With 34 billion parameters, it excels in code generation, code completion, and understanding programming languages, making it a valuable tool for software development and code-related research.

(5) **GPT-4-Turbo (gpt-4-0125-preview)**: GPT-4-Turbo is a highly optimized version of the GPT-4 model. It offers faster inference times and improved efficiency while maintaining the high-quality output that GPT-4 is known for. This model is particularly useful for applications requiring quick responses without compromising on quality.

(6) **Llama-3-8B-instruct**: Llama-3-8B is an iteration of the Llama model series, featuring 8 billion parameters. It provides a balance between computational efficiency and performance, making it suitable for a wide range of NLP applications.

(7) **Claude-3-Opus**: Claude-3-Opus is the latest in the Claude series, bringing substantial improvements in language understanding and generation. It integrates advanced techniques to enhance its contextual comprehension and generation capabilities, making it a top choice for sophisticated NLP tasks.

(8) **Llama-3-70B-instruct**: Llama-3-70B is a LLM with 70 billion parameters. This model is designed to tackle the most challenging NLP tasks, providing unparalleled performance in terms of accuracy and coherence in text generation.

(9) **GPT-4-32k**: GPT-4-32k is a variant of the GPT-4 model with an extended context window of 32,000 tokens. This extended context window allows it to handle long-form content more effectively, making it ideal for applications requiring extensive context retention and understanding.

## E.1   MODEL SHORTCOMING ANALYSIS

**Long-Context Challenges.**   The challenge of handling long-contexts is considerable in TAPILOT-CROSSING, especially for models with shorter maximum input lengths. Models such as Codellama-34B, which has a maximum input length of 16k, are particularly affected. For example, it is essential for LLMs to access all private function descriptions and codes for effective code generation with

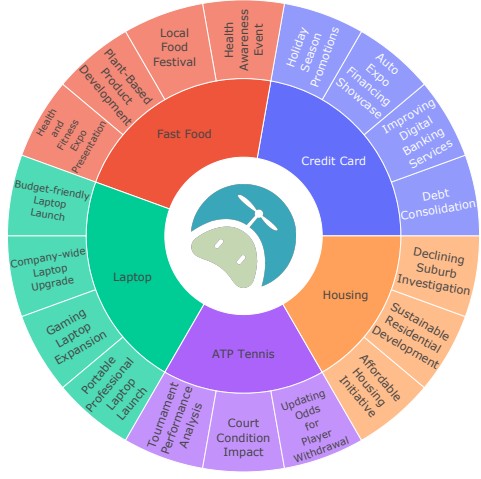
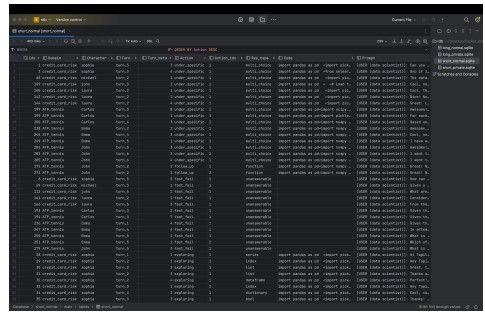

(a) Visualization of 18 topics and 5 data sources of TAPILOT-CROSSING.

(b) The screenshot of History Relational Database (H-RDB).

Figure 8: Visualization of 18 topics and 5 data sources and constructed H-RDB of TAPILOT-CROSSING.

retrieved functions. The statistics shows that the average number of prompt tokens for PRIVATE is 15.7k, and notably, 20.4% of their prompts surpass the 16k length.

**Instruction Following.** Our experiments reveal that GPT-4-32k requires minimal efforts in prompt design due to their exceptional ability to follow human instructions. To be specific, only 3.4% of their results deviate from the provided instructions. However, other models exhibit a higher proportion of unexpected result types. For instance, extracting generated codes or answers from Claude-2.1 proves to be extremely challenging since it often embeds the answer in the middle of outputs rather than at the end as defined. We also observe that GPT-4-Turbo tends to generate longer codes in any settings. While this characteristic enhances its performance in code generation, it also results in 60.3% of the code generated during ReAct reasoning being non-executable, thereby leading to incorrect answers. Furthermore, CodeLlama-34B-Instruct exhibits a lack of robustness when faced with longer or more complex prompts. With the addition of COT, the performance of CodeLlama significantly drops from 27.5% with simpler instructions to 18.5% in NORMAL code generation.

# F DYNAMIC HISTORY COMBINATION

## F.1 HISTORY RELATIONAL DATABASE (H-RDB)

From all the User-AI conversation data shown in Figure 8(a), we split the User-AI conversation into several single-turn user queries and AI answers stored in a relational database, indexed by the conversational order as shown in figure 8(b). This storage is subject to dynamic combinations for different scenarios.

## F.2 HISTORY RETRIEVAL QUERIES

When retrieving the stored history information, we use `sqlite3`[5] python package. The search query is provided in sqlite3 format, for example: `SELECT {Prompt} FROM {table} WHERE 1=1 AND Domain = ? AND ...`

---

[5]https://docs.python.org/3/library/sqlite3.html

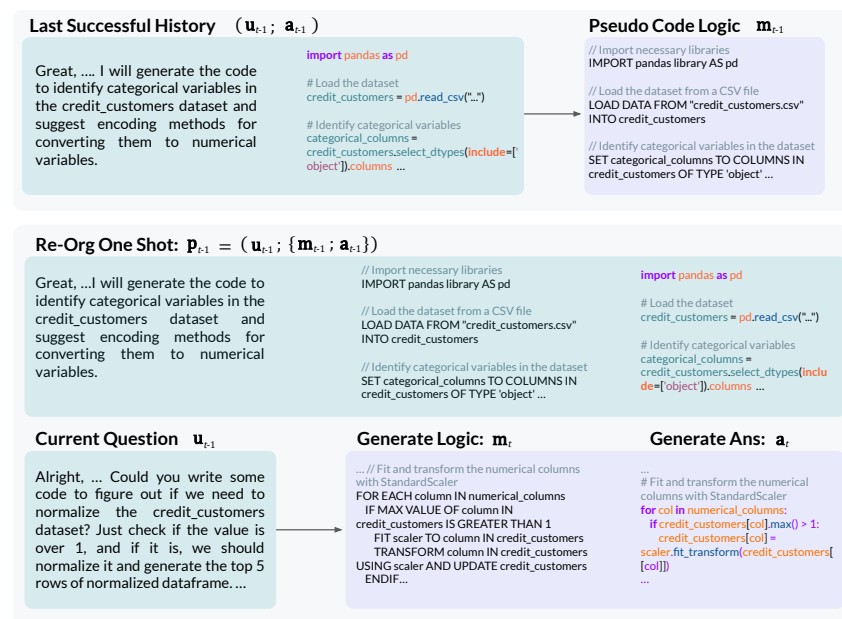

Figure 9: This is an overview of our proposed method, ACR. The areas highlighted in purple represent results generated by the agents.

### F.3 TAPILOT-ALPHA

As stated in Section R, the current TAPILOT-CROSSING involves only clean and accurate code revisions, which we refer to as the `Alpha` version. Looking ahead, we are considering the incorporation of noisy data or the integration of user conversations in `Action` mode into the code history. This potential expansion aims to simulate more realistic development environments and challenges. Even within the constraints of a curated and error-free conversation history, the experimental results show that there still are substantial opportunities for optimization and improvements.

## G DIALOG TYPES

TAPILOT-CROSSING can be categorized into **Statement-** (longer) and **Colloquial-** (shorter) dialogs. The statement-dialogs are more formal, resulting in more complex user instructions and code generations, which are commonly found in computational notebooks (Yin et al., 2023). On the other hand, colloquial dialogs involve shorter and simpler user questions, but exhibit more colloquial and conversational characteristics. This category of dialogs is primarily constructed through the process of prompting GPT-4 to segment and reinterpret the existing statement-dialogs.

## H ACR IMPLEMENTATION

Figure 9 presents the detailed steps of ACR.

## I AGENT IMPLEMENTATION

### I.1 MULTI-AGENT IMPLEMENTATION.

As a valuable and interesting agent type, Multi-Agent is recognized to have the potential to enhance performance in many reasoning tasks including BOLAA (Liu et al., 2023d) and MetaGPT (Hong

| Model | Normal | Conversation Mode | | Pri-Act | Answer Type | | Overall |
| | | Action | Private | | Code | Choice | |
|---|---|---|---|---|---|---|---|
| GPT-4-32k | 29.7 | 24.2 | 7.1 | 0.0 | 17.8 | 25.4 | 21.0 |
| + Agent | 23.4 | 39.2 | 9.1 | 5.3 | 16.6 | 38.8 | 25.9 |
| + Multi-Agent | **32.7** | 40.8 | 10.1 | 9.2 | 20.5 | **43.7** | 27.8 |
| + Inter-Agent | 32.2 | **41.3** | **10.6** | **9.8** | **21.6** | 42.1 | **30.2** |

Table 5: The results of GPT-4-32K in base, agent, Multi-Agent, and inter-agent modes on the TAPILOT-CROSSING dataset. **Pri-Act** refers to private library + action evaluation mode.

et al.). However, MetaGPT was designed specifically for software development requirements without mechanisms for handling structured data and conversation histories, making it less applicable to our problem setting. Therefore we implemented the Multi-Agent reasoning type as introduced in (Liu et al., 2023d), which is a more general framework and can be implemented more flexibly in different settings. To be specific, except central CONTROLLER, we also create TOOL AGENT, CODE AGENT, DECISION-MAKING AGENT, and PRIVATE-LIB AGENT. Our results in Table 5 clearly indicate that the Multi-Agent configuration obviously outperforms the original Agent setting and model base setting, underscoring its potential. Notably, it achieves performance on par with our Inter-Agent configuration, particularly showing improvement in Multi-Choice tasks due to the important role played by the DECISION-MAKING AGENT. This supports one of our motivations: beyond code generation, providing insightful analysis for users based on results is crucial in data analysis tasks. However, we note that the Multi-Agent requires higher costs in our dataset and more sophisticated prompt design for each agent. Moreover, its performance begins to decay with increasing turns since each agent must be provided with not only conversations with users but also conversations between agents. This results in a prompt token consumption that is approximately 5.3 times higher than that of the Inter-Agent. Therefore, this observation reinforces the necessity of our design of ACR in the Inter-Agent. It is not only effective but also efficient, so it's more suitable in conversational settings.

## I.2 TOOLKIT

**Executor.** To get the execution results of code generated by LLMs, we adopt Python Executor `exec()` which is implemented in Python [6], within a isolated Python environment. The output of the code execution, whether it be any return values, print statements, or error messages, is then captured by the Executor. This output is subsequently returned to the LMs, providing them with feedback on the results of their code generation to make a better next-step action or decision.

**User Simulator.** In addressing the clarification action type, LLMs are permitted to request clarification when they feel ambigous about conditions from user queires. Therefore, we employ GPT-4 Turbo (with fixed version) to emulate the question-answering behavior of users, considering that GPT-4 has been demonstrated to provide feedback of equivalent quality to human responses (Wang et al., 2024).

**Chart-to-Table.** We employ deplot (Liu et al., 2023a) to convert images into a table. Given the table, then LLMs can reason and answer the questions.

## I.3 REASONING

**COT.** To evaluate the pure code generalization capability of data analysis, we restrict LLMs from executing code during generation. Therefore, we employ a zero-shot COT for the reasoning of the `Agent` mode. The key prompt to implement such COT is:
```
...   write a step-by-step outline and then write the code:
```

**ReAct.** To evaluate analytical capabilities beyond mere code generation, we employ ReAct for multiple-choice questions. Specifically, we set the MAX STEP for ReAct reasoning to 5, with the Executor serving as the primary tool. Data analysis agents are tasked to generate, analyze, and draw

---

[6]`https://docs.python.org/3/library/functions.html#exec`

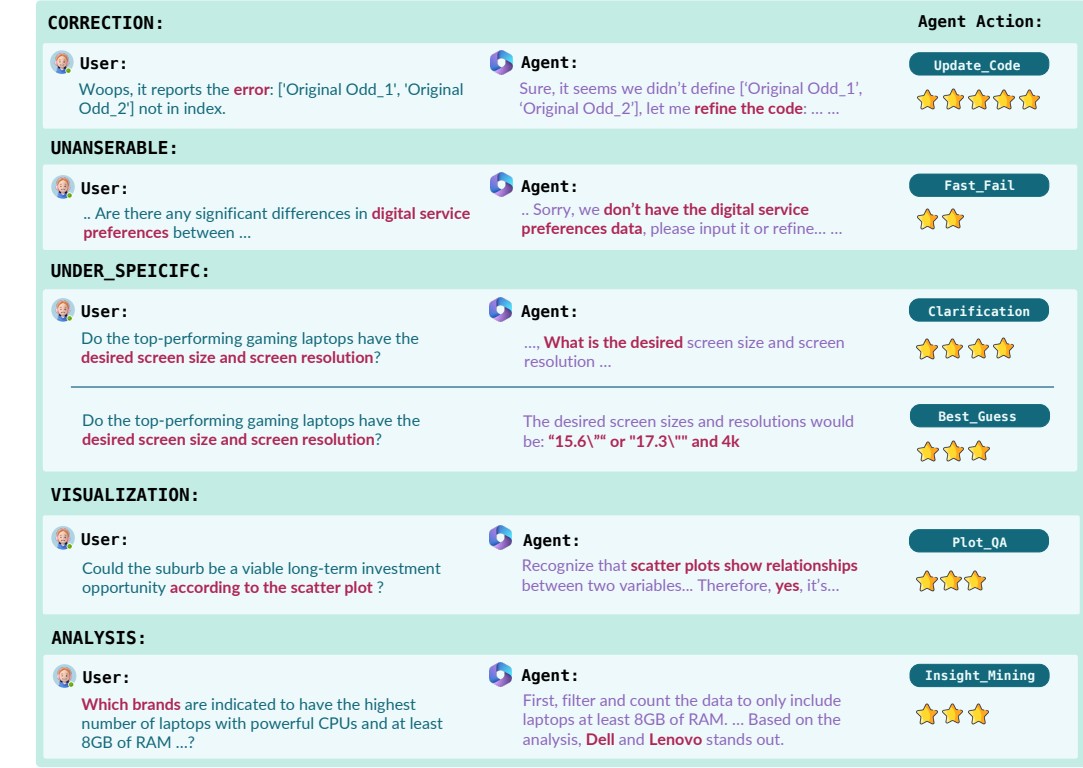

Figure 10: This figure provides an overview of action types in TAPILOT-CROSSING, illustrated by examples. We emphasize the keywords specific to each category, and demonstrate the relevant sections of the associated queries, as well as the agent actions. The number of ⭐ symbols represents the relative difficulty of each action. Please note that all free-text examples presented in this figure are only used for illustration purpose. **In TAPILOT-CROSSING, each answer format is limited to either code generation or multiple-choice questions.**

conclusions about their results. If the result contains bugs, the corresponding message is returned to the agent for rectification, although this process may consume additional reasoning steps. We also manually provide a one-shot example to guide agents on how to react in TAPILOT-CROSSING. To prevent data leakage, we cross-reference examples across different tabular data. For instance, an example curated from `ATP_Tennis` could be used to guide LLMs in the `Laptop Pricing` dataset.

## J    IMPLEMENTATION DETAILS

### J.1    GENERAL IMPLEMENTATION

The `temperature` parameter is set to 0.0 for Claude 2.1, GPT-4, and GPT-4-Turbo and `top_p` to 1.0.

## K    ACTION DESCRIPTION

In this section, we categorize and formalize the action types in TAPILOT-CROSSING, identifying five distinct sub-categories that correspond to different types of user queries.

### K.1    UPDATE_CODE

The `Update_Code` action refers to instances where the user requests corrections for bugs or refinements to the conditions of previous queries.

### K.2 FAST_FAIL

`Fast_Fail` is an action that alerts users when the current data contents or resources are insufficient to meet their requests, or when user queries contain factual errors.

### K.3 CLARIFICATION

`Clarification` is a common action in response to under-specified questions, which are frequent in data-analysis queries. In this action, agents make the conditions of the question more specific and clear by seeking additional information from users.

### K.4 BEST_GUESS

While `Clarification` is an effective action to reduce the uncertainty, it can lead to issues such as user impatience due to unsteadily asking, and long dialog histories that result in attention distraction and long-context problems. Therefore, the `Best_Guess` action can address these issues by making appropriate assumptions based on data contents, domain knowledge, and commonsense knowledge for under-specific questions. However, there is also a risk that incorrect guesses can lead to hallucinations.

### K.5 PLOT_QA

In real data analysis settings, agents are also expected to answer user questions about insights derived from plots. The `Plot_QA` action can assist users in better understanding the contents of plots for decision making.

### K.6 INSIGHT_MINING

Beyond generating codes for users to retrieve expected results, code agents are also tasked with summarizing executed results from the environment to assist users in making informed decisions. This process, known as `Insight_Mining`, plays an important role in data analysis since it contributes to the evolution of code agents into comprehensive data analysis agents.

## L EVALUATION METRIC DETAILS

We introduce evaluation metric details in Section L.1, and implementations for each result type. And the distribution of result typs is presented in Figure 11(b)

### L.1 EVALUATION METRICS

**Accuracy (Acc).** Acc is a common metric that evaluates ability of agents to generate code that executes correctly or to accurately answer multi-choice questions. It is defined as the proportion of instances where the predicted outputs match the expected reference output, across all evaluated tasks. For a given dataset with $N$ instances, where $C_i$ is the expected outcome (either execution result or correct answer) and $\hat{C}_i$ is the predicted outputs for the $i^{th}$ instance, Acc is calculated as follows:

$$\text{Acc} = \frac{1}{N} \sum_{i=1}^{N} \mathbf{I}(C_i = \hat{C}_i), \tag{1}$$

where $\mathbf{I}$ is an indicator function that returns 1 if $C_i = \hat{C}_i$, and 0 otherwise.

**Acc with Private Lib Recall (AccR).** Recognizing the importance of accurately leveraging specific user-defined libraries in code generation, we extend Acc to include a recall-based adjustment for instances involving private libraries. This ensures that AccR not only evaluates the direct accuracy of code execution and question answering but also evaluates the inclusion and correct usage of private library functions. AccR can be computed as follows:

$$\text{AccR} = \frac{1}{N} \sum_{i=1}^{N} \mathbf{I}(C_i = \hat{C}_i) \cdot \mathbf{R}(C_i, \hat{C}_i), \tag{2}$$

$$\mathbf{R}(Ci, \hat{C}i) = \frac{|\mathbf{F}(Ci) \cap \mathbf{F}(\hat{C}i)|}{|\mathbf{F}(Ci)|}, \tag{3}$$

where $\mathbf{R}(C_i, \hat{C}i)$ quantifies the recall rate of relevant library functions in the predicted code. $\mathbf{F}(C_i)$ and $\mathbf{F}(\hat{C}_i)$ denote the set of private library functions in the reference codes and the set actually utilized by agents in the predicted codes, respectively. The final score would be weighted sum of Acc and AccR.

## L.2 DATAFRAME COMPARISON

The function compares two dataframes (`df_1` and `df_2`) by checking their indices, column presence, and column data. It uses `np.allclose()` for numeric data and direct comparison for non-numeric data. If a column in `df_1` is absent in the original dataframe, it searches for a matching column in `df_2`. The function returns `True` if `df_1` and `df_2` are equivalent, otherwise `False`. Please note, the `column names` will not be computed since different LLMs may have their only preference names. For example, the `win_ratio` generated by GPT-4 could be called `winning ratio` by Claude 2.1.

## L.3 VISUALIZATION COMPARISON

We note that it is hard to compare the closed-form results for visualization-based code generation since parameters of plots may be varied significant across models. For instance, GPT-4 generated plots may be the same with CodeLlama while their title names may be different, which leads to false negatives. Therefore we utilize `PIL` package to compute similarity between plots. To be specific, the function `compare_plots` takes two image file paths as inputs (`ai_output` and `reference_output`), resizes them to 800x600 pixels using the `LANCZOS` method, and saves them. The images are then read in grayscale mode to avoid the difference brought by colors. The function computes and returns the Structural Similarity Index (SSIM), a measure of image similarity, between the two images. This function can be used to compare an AI model's output with a reference output. Finally, the code generated will be considered as correct if the similarity is larger than 0.6.

## L.4 MULTI-INTENT EVALUATION

In this work, we evaluate the code generation performance on intent manner, which means if one user query contains multiple intents, then the total scores of this query will be the number of intents. We evaluate each intent separately and sum up the scores of all intents as the denominator when calculate the performance of each model in percentage.

## L.5 PRIVATE FUNCTION RECALL

We notice that some LLMs tend to import as many as possible private functions while not using all of them. Thus, to extract all indeed used private functions in the customized function library, we utilize `AST` package. After extracting the used private functions, we calculate the recall coefficient according to Equation 3.

## L.6 CODE SIMILARITY EQUIVALANCE (CSE)

In the context of TAPILOT-CROSSING, the complexity of code generation tasks—many of which yield a score of zero—presents huge challenges in evaluating performance through Acc or AccR only. This is particularly evident when distinguishing between codes that differ by merely a single line of error or output, both of which would result in an Acc or AccR of zero, despite their obvious differences in code generation capabilities. To overcome this limitation, we propose the introduction of Code Similarity Equivalence (CSE), a nuanced evaluation metric designed to assess the similarity between generated codes and reference codes. Given that these codes originate based on identical user instructions, a high degree of similarity is expected. Our approach leverages a hybrid combination of models to reduce the bias, incorporating CodeT5++ and OpenAI Ada (`text-embedding-ada-002`) models, which are affordable and available for most institutes. This combination has demonstrated a strong correlation with human evaluative preferences, offering a more refined and accurate measure of code generation performance.

**Details.** We introduce here about how to conduct more nuanced evaluation of Acc or AccR with CSE. 1) We collect 180 instances of code generation including both NORMAL and PRIVATE. To evaluation the quality of these codes, we enlist two additional PhD students who are proficient in data science and Python as evaluation committee.

2) They evaluate code generated by several models, including GPT-4-32k, GPT-4-Turbo, Claude-2.1, CodeLlama-Instruct (ranging from 7B to 34B parameters), StarCoder, and DeepSeek-Coder-Instruct (also from 7B to 34B parameters). Each evaluator is provided with comprehensive user code histories, tabular contents, the current query, access to the `decision_company` private library. Please note that evaluations are conducted only based on their expertise and experience, without any predefined guidelines and discussion, to avoid bias.

3) We ask for a relative ranking of generated codes among models over absolute scoring to avoid potential variability in scoring preferences among the evaluators.

4) In cases of parts of divergent rankings, the evaluators engage in discussions regarding the specific code samples until a consensus was reached. This step ensures a more reliable and agreed-upon evaluation outcome.

5) The evaluation committee then examine various open-source and readily available embedding models to measure code similarity, aiming to closely match their ranking preferences. Our exploration identifies that the score system consisting of CodeT5+ (Wang et al., 2023b) and Ada (`text-embedding-ada-002`) most closely aligned with human evaluative preferences.

**Introduction of a Mixed Evaluation Metric (AccSE & AccSER).** To accurately reflect the nuanced capabilities of code generation models, we propose a composite metric that integrates Code Similarity Evaluation (CSE) with Accuracy (Acc), termed Accuracy for Similarity Evaluation (AccSE). This metric is concisely defined as:

$$\text{AccSE} = \begin{cases} 1.0, & \text{if } C = \hat{C}, \\ 0.5, & \text{if } S_1 > 0.85 \wedge S_2 > 0.9, \\ 0.25, & \text{if } (S_1 > 0.85 \wedge S_2 \leq 0.9) \\ & \vee (S_1 \leq 0.85 \wedge S_2 > 0.9), \\ 0, & \text{otherwise.} \end{cases} \tag{4}$$

Where:

- $C$ and $\hat{C}$ represent the reference and generated code execution outcomes, respectively.
- $S_1$ denotes the CSE score based on CodeT5+.
- $S_2$ denotes the CSE score based on Ada.

This formulation succinctly captures the evaluation criteria for AccSE, with symbols $S_1$ and $S_2$ representing the CSE scores based on CodeT5+ and Ada, respectively. The logical operators $\wedge$ and $\vee$ are used for "and" and "or" conditions, respectively, to further compact the notation. AccSER is computed in the similar way just times recall score for each value as Eq. 3.

We hold this for future evaluation system of TAPILOT-CROSSING when we conduct more extensive cases with involved with more expert volunteers.

**Rationale Against GPT-4-Based and Multi-Agent Evaluation Methods.** While existing research suggests that GPT-4-based soft evaluation could enhance the assessment of complex generative tasks, such approaches are deemed unsuitable for TAPILOT-CROSSING due to several critical reasons:

1) **Bias Concerns:** The prototype annotations and questions in our study originate from a GPT-4-based agent environment. Employing GPT-4 for evaluation purposes could inadvertently introduce a self-enhancement bias (Zheng et al., 2024a), compromising fairness across model evaluations.

2) **Cost Concerns:** Although multi-agent evaluation frameworks, incorporating diverse families of LLMs, is to mitigate bias (Li et al., 2023c), the economical and computational overhead is obvious. Specifically, evaluations in such settings require at least twice the token consumption than that used in generation alone, rendering it impractically expensive in TAPILOT-CROSSING.

Given these considerations, our research proposes an alternative evaluation methodology that is both cost-effective and reliable for evaluating the accuracy of complex data science code generation at this

time. We demonstrate that CodeT5+, a remarkably efficient code embedding model, can obviously distinguish between varying performance levels and accurately identify correct code logic. Crucially, this model offers a pragmatic balance between evaluation thoroughness and resource efficiency.

### L.7 OTHER VALUE TYPES

For other result types, such as dictionry, set, list, we directly compute the exectued results and determine whether they are equal or not.

### L.8 CASE-BY-CASE EVALUATION

While we categorize instances according to result types and provide evaluation codes for each type, some scenarios requires a case-by-case evaluation script. For instance, in most dataframe or matrix comparisons, we employ `np.close()` and `string` match for result comparison. However, in some cases, such as using a dataframe or matrix to display a classifier's Confusion Matrix, the predicted code is deemed correct if its `f1-score` surpasses that of the referenced code, even if their `f1-scores` are not similar. For the evaluation script of TAPILOT-CROSSING, we manually review and adjust the scripts to accommodate each case.

## M ACTION EVALUATION MODE

### M.1 CORRECTION

**Update_Code.** This could be evaluated within a static setting where the bug feedback is embedded into user-code history. Agents are requried to update the previous code via user feedback.

### M.2 UNAWSERABLE

**Fast_Fail.** In DECISION COMPANY, we keep the original unanswerable questions and categorize them as multi-choice questions. This is done to evaluate if agents can identify these questions based on their analysis of table contents and commonsense knowledge. To prevent any biased setting, such as specially designed prompts that might mislead agents into determining a question as unanswerable, we sample an equal number of under-specified problems and answerable questions. We then reformulate their choices, enabling the model to decide whether a question is answerable with clarification or assumption, or to directly classify it as unanswerable.

### M.3 UNDER_SPECIFIC

**Clarification.** To evaluate the performance of agents on clarification action, we employ a dynamic setting that incorporates a User Simulator. This simulator mimics user feedback based on the ground truth code or answer. Initially, conversational data analysis agents are expected to pose questions for clarification, simulator will answer it according to the ground truth answers. Subsequently, these agents are tasked with generating the final code, understanding both the original history and the history of clarifications. This setup provides a robust assessment of the agents' ability to converse with human, clarify ambiguities, and generate accurate code.

**Best_Guess.** We aim to evaluate the ability of conversational data analysis agents to make accurate assumptions when faced with ambiguous questions, without resorting to constant clarification, which could potentially frustrate users. We believe that the best guess of an agent should not impact the final decision and this evaluation metric should be somehow flexible. For instance, in a credit card application scenario, the term `young people` could refer to individuals aged 20-40 or 25-45, making it challenging to be evaluated by fixed metrics. Therefore, we opt to use multiple-choice questions to assess the agents' assumption-making capabilities. We posit that an assumption is appropriate only if it does not influence the final decision-making process.

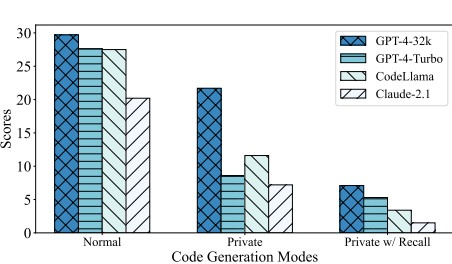
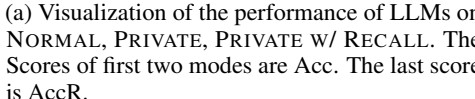

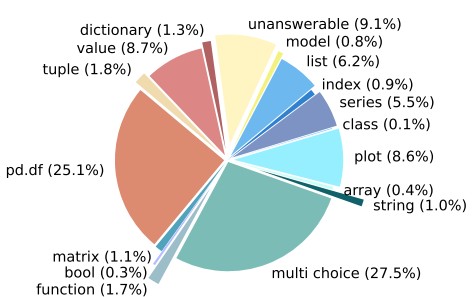

(a) Visualization of the performance of LLMs on NORMAL, PRIVATE, PRIVATE W/ RECALL. The Scores of first two modes are Acc. The last score is AccR.

(b) Visualization of the results types distribution of TAPILOT-CROSSING.

Figure 11: Visualization of the performance of LLMs with PRIVATE mode and results types distribution.

### M.4 VISUALIZATION

**Plot_QA.** We evaluate the analysis capability of agents around plot in TAPILOT-CROSSING. The end format of answer would be multiple choices.

### M.5 ANALYSIS

**Insight_Mining.** We evaluate the analysis capability of agents generally in TAPILOT-CROSSING. We opt to use multi-choice questions to evaluate it.

## N  PRIVATE MODE ANALYSIS

**Overall Results.** Table 3 and Figure 11(a) indicates that the PRIVATE setting presents a considerable obstacle, with the best performing GPT-4-Turbo `Inter-Agent` only achieving 12%. This demonstrates that understanding and implementing user-specific functions is a critical and urgent skill for LLM agents in real-world data analysis tasks (Zan et al., 2022).

**The Critical Role of Function Relative Recall.** Notably, CodeLlama outperforms GPT-4-Turbo in Acc within the PRIVATE setting. However, its performance declines significantly relative to GPT-4-Turbo upon the consideration of private library relative recall in the generated codes, as measured by the AccR metric. This observation suggests that CodeLlama tends to reply less on user-defined private functions, aiming to reduce risk of code errors. Therefore, AccR metric can spotlight the balance required between proficient code generation and the meticulous integration of user-specified private libraries to foster safer and more satisfying code production.

## O  DETAILED ERROR ANALYSIS

In this study, the error patterns exhibited by tested LLMs are critically examined to provide insight into the predominant challenges faced during their operation. A detailed discussion is provided in Table 6. We analyze 200 randomly sampled instances, categorizing errors into three main types as follows: **(1) Key Error (21%)** occurs when the model incorrectly assumes the existence of a column name in the provided data table which does not exist. This error reflects a fundamental misinterpretation of the table information and hallucination where the model uses non-existent fields for data retrieval. An example is the model's incorrect use of '`high_credit_long_duration`['employment_duration']' instead of the correct attribute '['employment']'. This error type suggests that the model may overly rely on its trained patterns rather than accurately assessing the real structure of the data, leading to 'hallucinated' column references. **(2) Lazy Assumption (37%)** refers to instances where the model tends to assume that intermediate results or states are already available or saved on disk.

This often leads to erroneous or incomplete code execution paths, such as the premature use of 'updated_odds_df' without ensuring its prior creation and calculation. This type of error may arise because models often seek shortcuts in their processing, opting to retrieve and manipulate existing objects rather than generating solutions from scratch. This tendency can reduce the reliability and flexibility of the model, as it may fail under conditions where dependencies are not pre-established. **(3) Bad Instruction Following (56%)** describes the model's failure to adhere strictly to given instructions, resulting in an inability to properly answer the posed questions. This is exemplified by the response of the model with 'D. None of above' when asked to generate Python code to help solve a question, showing a lack of direct engagement with the query requirements. This type of error often becomes more pronounced in later conversation turns, suggesting a compounding of misunderstandings or a degradation of context over the course of a session. These error types are critical in understanding the limitations of the current model, guiding future improvements in LLM agents' evolving abilities. Understanding these patterns helps in pinpointing specific areas where training data, model architecture, or conversation protocols need enhancement to improve overall performance and reliability.

**Error Type Analysis Across Different LLMs and Conversation Settings.** Figure 7 showcases the distribution of error types across various LLMs and settings. This figure provides a comparative insight into the frequency of three primary error types: Key Error, Lazy Assumption, and Bad Instruction Following, both in Model-Base and Inter-Agent setting. **Key Error** rates vary significantly across models. For instance, Claude-2 exhibits a notably higher rate of Key Errors compared to other models, which might suggest a less effective understanding or integration of database schema information in this model. **Lazy Assumption** errors are consistently high across all models, indicating a common model behavior where assumptions are made about the state of computations or data availability. This could reflect an inherent model optimization to minimize computational expense by reusing existing data states or structures, which, while efficient, can lead to inaccuracies when those states are not correctly initialized or updated. **Bad Instruction Following** shows a general high trend across models, particularly noticeable in settings involving Inter-Agent conversations. This suggests that as models engage in more complex dialogues or tasks requiring cooperative problem solving, their ability to follow detailed instructions without deviation diminishes. This could be due to accumulating contextual misunderstandings or the increasing complexity of managing multiple instruction streams. **Inter-Agent Variations** are particularly interesting; while Key Errors and Lazy Assumptions increase slightly, Bad Instruction Following errors show a marked increase. This may be due to the added complexity of coordinating and maintaining consistent task strategies between agents, highlighting a critical area for further research and model training refinement. These insights are crucial for understanding specific weaknesses in current LLM implementations and point towards necessary areas for improvement in model training protocols and architecture adjustments. Enhanced training methods focusing on schema understanding and multi-agent coordination could mitigate some of these prevalent errors.

# P    MORE STUDY

## P.1    LARGE LANGUAGE MODELS FOR DATA ANALYSIS

The use of LLMs for data analysis has been a topic of interest in recent years. LLMs powered by In-Context Learning (Yang et al., 2023; Dai et al., 2023; Dong et al., 2023) have been employed in various data analysis tasks, such as SQL query generation (Pourreza & Rafiei, 2024a; Gao et al., 2023; Lei et al., 2023; Zhang et al., 2023b; Gu et al., 2024; Wang et al., 2023a; Pourreza & Rafiei, 2024b; Li et al., 2024b), pandas or python code generation (Jain et al., 2023; Chen et al., 2024; 2023a; Li et al., 2024c; Zha et al., 2023; Zhang et al., 2023a; Zheng et al., 2024b), and data visualization (Chen et al., 2023b; Huang et al., 2023a). However, most of these works focus on single-turn setting, where the user query is explicit and does not require any conversation or clarification. Recently, there has been a growing interest in conversational data analysis, where the user intents may need to be clarified or refined through conversational communication (De Vries et al., 2020; Yan et al., 2023; Wang et al., 2024).

| Error Type | Definition | Example |
|---|---|---|
| Key Error (21%) | Refers to the instance where the model imagines a reasonable but non-existent column name to retrieve the data table. | **Question:** *We want to find clients who have stable employment. We can consider stable employment as those with employment duration of 4 years or more.*
**Gold:** high_credit_long_duration[high_credit_long_duration[**'employment'**] == 'x>=4']
**Error:** high_credit_long_duration[high_credit_long_duration[**'employment_duration'**] == 'x>=4'] |
| Lazy Assumption (37%) | Refers to the instance where the model tends to assume some middle results have already been prepared. | **Question:** *Now, I need to know the potential impact of the updated odds on the later rounds of the tournament.*
**Error:** # Assuming **'updated_odds_df' is already created** and contains the updated odds
updated_odds_df = pickle.load('updated_odds_df.pkl') |
| Bad Instruction Following (56%) | Refers to the instance where the model can not follow instructions well, leading to the failure of answering questions. | **Question:** *Please answer the multi-choice question ... ...; you will need to generate Python code which can assist yourself to answer the question in this step.*
**Error:** **'D. None of above'**
Explanation: There is no specific test or condition to check ... ... |

Table 6: Definitions and Examples of main error types.

## P.2 Data Analysis Benchmarks

The development of benchmarks for data analysis tasks has been a crucial factor in driving the progress of LLMs in data science. Existing benchmarks can be broadly categorized into single-turn and multi-turn benchmarks. Single-turn benchmarks, such as HumanEval (Chen et al., 2021), MBPP (Austin et al., 2021), Spider (Yu et al., 2018), BIRD (Li et al., 2023a), Text2Analysis (He et al., 2023), DABench (Hu et al., 2024) and DS-1000 (Lai et al., 2023), focus on generating code snippets or closed-form insight summaries for data analysis given a single user query. To explore conversational nature of real-world data analysis scenarios, where the user intent may need to be clarified or refined through conversational communication, several multi-turn benchmarks have been proposed, including CoSQL (Yu et al., 2019a), and ARCADE (Yin et al., 2023). However, these benchmarks are primarily focused on code generation and do not cover other aspects of data analysis, such as data visualization and understanding based on intermediate results. Our work extends the existing literature by introducing a new benchmark, TAPILOT-CROSSING, for evaluating LLM agents in conversational data analysis tasks across wide range of data analysis settings.

## P.3 Multi-Agent Environments for Data Generation

LLMs have proven to be effective in constructing multi-agent environments for automatic data generation. For instance, Lu et al. (2023) and Ding et al. (2023) simulate dialogs for QA and text generation tasks. Also Li et al. (2023b) generates data about API calls using multi-agent environments. This is because LLM agents can simulate believable human actions when placed in an environment with dynamically updating knowledge and memory (Park et al., 2023). Inspired by this, we also created DECISION COMPANY to generate conversation log data for data analysis with more believable behaviors. Unlike most of previous work on training dataset generation, our research pioneers the construction of the conversational benchmark with a specific focus on conversational data analysis agent evaluation.

## Q Dataset Quality Evaluation Details

### Q.1 Human Evaluation

To evaluate the quality of the dataset annotation, we also conduct a thorough human evaluation. We select 500 samples and invite 10 experts with extensive data analysis experience to review the dataset. The evaluation metrics listed below are carried out using a binary scoring system, with scores of 0 or 1 (Reject or Accept). We divide the measurement metrics into two levels: **General Metrics** and **Action-wise Metrics**, which are elaborated below.

**General Metrics.** To ensure the overall quality of the dataset, we apply a set of comprehensive general metrics. These metrics are designed to evaluate the ability of dataset to capture meaningful, diverse, and coherent multi-turn conversations in the data analysis domain. Here is a brief introduction:

- **Conversation Coherence:** Evaluate whether the dataset contains logically consistent conversations including generated codes that flow naturally across multiple turns and lead to expected answers.

- **Scenarios Diversity and Reasonableness** Assess whether the dataset contains a wide range of scenarios without duplication, tasks, and user intents. And whether they are reasonable and can be fully supported by the given tabular data. This metric is satisfied if both sub-metrics are satisfied.

- **Conversation Topic Coherence:** Measure the overall relevance of the conversation to the given topic, ensuring that the conversation stays on track and go off the data analysis questions. This metric is satisfied if both sub-metrics are satisfied:

  - Conversation Goal Relevance: Evaluate whether each conversation turn contributes meaningfully to achieving the final goal. Turns must remain focused on the final conversation goal.
  - Table Relevance: Measure whether at least 40% of the conversation turns involve conversation with the provided tabular data for specific conditions. This ensures the conversation is sufficiently relevant to the dataset being analyzed.

- **Ethics and Bias Representation:** Assess whether the dataset avoids biased, harmful, or unethical content.

- **Conversation Naturalness:** Measure whether the conversation in the dataset reflect natural, conversational language, avoiding overly robotic or AI-like responses. Conversations should resemble real human conversations in tone and flow.

- **Evaluation Scripts Quality:** This metric is satisfied if all sub-metrics are satisfied.

  - Tool Reliability: Measure whether tools usage logs such as the logs of user simulator tool are reasonable and trustworthy, and whether the Python executor tool utilizes common, reliable packages to ensure consistent and accurate results.
  - Evaluation Script Flexibility and Comprehensiveness:
    1. Flexibility: Ensure that the evaluation scripts are not overly rigid and can accept multiple valid, reasonable outputs as correct, allowing for variations in agent responses.
    2. Comprehensiveness: Measure whether the scripts are robust enough to handle a wide range of scenarios, including corner cases, ensuring they effectively evaluate all possible outcomes.
    3. For multi-choice questions, the provided options should be valuable and challenging enough, where can't be easily figured out from merely question and options.

- **Evaluation Script Scalability:** Measure how easily the evaluation scripts can be extended or adapted to accommodate new data. This metric evaluates whether the framework allows for seamless integration of new evaluation scripts without requiring significant modifications, ensuring efficient scalability as the dataset grows.

**Action-wise Metrics.** In addition to the general metrics, we apply specific metrics to evaluate the detailed actions captured in the dataset. These action types include `Update_Code`, `Fast_Fail`, `Clarification`, `Best_Guess`, `Plot_QA`, and `Insight_Mining`.

Each action type is evaluated using core metrics to ensure its relevance, accuracy, and contextual consistency within the dataset. These metrics guarantee that the dataset reflects realistic and valid scenarios in the data analysis domain. Here is an overview:

- **Action Commonness in Data Analysis:** Evaluate whether the action cases are commonly seen in real-world data analysis tasks. This ensures that the dataset captures authentic, practical scenarios that analysts frequently encounter.

- **Correctness of Reference Answers:** Assess whether the ground-truth or reference answers provided in the dataset are accurate. The dataset should provide correct, verifiable solutions to the user query represented in each action.
- **Contextual Reasonableness:** Measure whether the context surrounding the action is comprehensive and logical. This ensures that the action occurs within a reasonable, coherent dialogue flow, taking into account all relevant factors from previous turns.

**Dataset Quality Statistics.** We compare the quality of the 500 sampled annotated data before and after human calibration, where the pre-calibration data was fully annotated by large language models (LLMs). As shown in Table 2, there is an obvious improvement in dataset quality following human calibration performed by two PhD students. The low acceptance ratio of the data prior to calibration underscores the necessity of this process. After calibration, the acceptance ratio rises to approximately 0.95, indicating that the involvement of two PhD students who are experienced in data analysis is sufficient to ensure the quality of the dataset, thus demonstrating the trade-off between efficiency and quality of our annotation workflow.

Specifically, the metrics for Scenario Diversity and Reasonableness improved from 0.46 to 0.96, reflecting enhanced coverage and variety of scenarios as shown in Section 3.1. Moreover, Conversation Topic Coherence increased from 0.17 to 0.93, indicating better alignment with the conversation topics. The Ethics and Bias Representation metric achieved a perfect score of 1.00, confirming the adherence of the dataset to ethical standards, while Conversation Naturalness improved from 0.67 to 0.95, suggesting more natural, human-like dialogues. Additionally, our Evaluation Scripts Quality and Evaluation Scripts Scalability scoring 0.98 and 0.94, respectively, prove an efficient solution to be against with data leakage problems. Overall, these results highlight the effectiveness of human calibration in enhancing both the conversation quality and the robustness of DESCISION COMPANY and high quality of TAPILOT-CROSSING.

Human evaluation is also conducted with a focus on the actions. Figure 5 illustrates the consensus that all actions in TAPILOT-CROSSING are both necessary and commonly observed in real-world data analysis scenarios. Furthermore, our simulated scenarios successfully capture and reflect key characteristics of these real-world conversational data analysis conversations.

| Category | pandas | matplot | Machine Learning (sklearn, scipy, seaborn) | numpy |
|---|---|---|---|---|
| **Percentage (%)** | 56.88% | 8.02% | 16.17% | 12.02% |

Table 7: Package Diversity of Our Dataset

### Q.2 DATASET DIVERSITY

We acknowledge the importance of data diversity and believe our benchmark, TAPILOT-CROSSING, effectively demonstrates it across multiple dimensions. We measure data diversity through several aspects:

- Domain / Topic Diversity, shown in Figure 8(a).
- Result Type Diversity, covering a wide range of query types requiring different code techniques, as displayed in Figure 11(b).
- Action Diversity, presented in Figure 4, where we demonstrate the comprehensive coverage of various action types in conversational data analysis. To our knowledge, we are the first to offer such extensive action type diversity in this domain.

Additionally, we evaluate data diversity through two more specific metrics:

- Package Diversity: Table 7 shows the distribution of query topics that cover various Python packages commonly used in data analysis, such as pandas, matplotlib, and machine learning libraries (sklearn, scipy, seaborn), with pandas dominating at 56.88%, followed by machine learning packages (16.17%) and numpy (12.02%).
- Query Diversity: Following the methodology in (Li et al., 2023a), we compute n-grams ($n = 3$) to reflect the diversity of each query. We compare TAPILOT-CROSSING against

| Metric | Tapilot-Crossing (1024) | DS-1000 |
|---|---|---|
| Average unique 3-grams in reference_answer (including codes & multi-choice answers) | 46.89 | 11.63 |
| Average unique 3-grams in current_query | 104.22 | 115.31 |
| Average unique 3-grams in prompt_with_context (history) | 1131.79 | 115.31 |

Table 8: Average Unique 3-grams

| Metric | Tapilot-Crossing (1024) | DS-1000 |
|---|---|---|
| Total unique 3-grams in reference_answer | 47,777 | 11,633 |
| Total unique 3-grams in current_query | 85,582 | 115,305 |
| Total unique 3-grams in prompt_with_context (history) | 1,046,955 | 115,305 |

Table 9: Total Unique 3-grams Counts

DS-1000 (Lai et al., 2023), a popular data analysis benchmark, to show the diversity of queries in our dataset.

- Average Unique 3-grams: Table 8 illustrates that TAPILOT-CROSSING provides a much higher diversity in reference answers, averaging 46.89 unique 3-grams compared to 11.63 of DS-1000. For current queries, TAPILOT-CROSSING maintains a comparable level of diversity, but when context is considered (prompt_with_context), TAPILOT-CROSSING vastly outperforms DS-1000, demonstrating its ability to handle complex, context-driven conversations with an average of 1131.79 unique 3-grams.

- Total Unique 3-grams Count: It show the diversity of the whole dataset. Table 9 further supports the diversity of TAPILOT-CROSSING, with the total number of unique 3-grams in reference answers reaching 47,777, significantly surpassing 11,633 of DS-1000. The total unique 3-grams in current queries stand at 85,582, and in prompt_with_context, TAPILOT-CROSSING achieves an outstanding 1,046,955 unique 3-grams, compared to 115,305 of DS-1000. This further highlights the rich contextual conversations captured by TAPILOT-CROSSING.

Overall, these metrics, along with the diversity in domains, result types, and actions, underscore extensive coverage of real-world data analysis tasks in TAPILOT-CROSSING. It surpasses existing benchmarks in capturing the full range of complexity and diversity needed for robust evaluation of conversational data analysis.

## R  LIMITATIONS AND FUTURE WORK

**Dataset Limitations.**  1) The TAPILOT-CROSSING dataset assumes that all human-machine conversation history is clean and correct. However, in real-world scenarios, the conversation history is often not clean, and may contain noise or require multi-turn clarifications for a single question. Therefore, future work should consider a more realistic, noisy conversational benchmark in data analysis. It is also worth noting that even with a clean history, the most capable model, GPT-4-32k, only achieves a score of 30.2 in the `Inter-Agent` mode. 2) The creation of our dataset is both cost-effective and efficient; however, the evaluation phase demands considerable efforts due to the inherent complexity and unpredictability of data analysis questions. Given it is challenging to discern subtle differences in performance among data analysis agents, especially in the context of long-form code generation and execution accuracy. This difficulty is exacerbated by the fact that the execution results of code with a single error (i.e., one-line error) and a completely incorrect code (just one-line output) are also determined as 0. Therefore, a soft-metric evaluation system should be introduced in the future as Appendix L.6. It would improve our ability to accurately gauge how close an answer is to the expected output, even when the executed output is zero, thereby providing a more fine-grained observation of code generation capabilities. 3) Finally, our work only concentrate on tabular data based analysis, while in the future, we would like to involve relational database (RDB)-based analysis with the programming language of SQLs.

**Method Limitations.** Our proposed reflection strategy can evolve LLMs into more effective conversational data analysis agents, but relies heavily on the accuracy of previous conversations. This reliance becomes less reliable in instances where the historical dialogue is cluttered with errors, suggesting the need for retrieval-augmented tools or methods to identify successful past conversations. Additionally, this strategy does not enhance agent performance in initial conversations due to the absence of historical data. Finally, while effective in many ACTION settings, this focus on conversation history may limit the inferential capabilities of LLMs by prioritizing past conversations over present context. Future efforts will be directed towards refining this approach to better balance the benefits of leveraging historical conversations against the need to maintain or enhance the inferential capabilities of LLMs.

## S  ETHICAL STATEMENT

The application of LLMs for automatic data generation requires a rigorous examination of ethical implications. The primary concern is the potential for LLMs to generate contents that could be considered harmful or biased. To mitigate these risks, human annotators (two PhD students) already filter and fix all problematic cases in Section 3.2. Also, LLMs may disseminate private or sensitive information. Therefore, we employ anonymization techniques wherein personal identifiers are systematically altered. For example, the name strings are replaced randomly, and any information of personas are switched as well. And the geographical locations of `John Smith` will be replaced with locations of `Carlos Garcia` to prevent any linkage to real-world individuals or entities. These procedures are conducted in Section 3. Moreover, we are committed to ensuring that the outputs generated by our LLM, referred to as TAPILOT-CROSSING, are free from political or sexual biases. To this end, each output, including conclusions and generated responses, is rigorously reviewed by the authors. In a nutshell, our ethical framework is built on a foundation of transparency, accountability, and a proactive stance towards mitigating any ethical concerns associated with the use of LLMs. The measures we have implemented reflect our commitment to upholding the highest standards of ethical research practice with LLMs

## T  DECISION COMPANY PROMPT

The process begins with the generation of client personas, as shown in Figure 12, where the Administrator agent is prompted to create meaningful personas. Following this, we simulate diverse analysis scenarios using In-Context Learning (ICL), which is depicted in Figure 13, allowing us to explore a wide range of potential outcomes. A critical aspect of the system is the discussion of analysis plans, where the conversation between the Data Scientist agent and the Client agent, illustrated in Figure 14, results in the generation of a series of analysis plans. To further support the process, conversation logs are annotated to capture the essence of conversations, with Figures 15 and 16 showing the perspectives of the Data Scientist Agent and the AI Chatbot Agent, respectively. Lastly, the evolution of our private library is detailed in Figure 17, which demonstrates the framework for prompting GPT-4 to generate code automatically, while human intervention plays a key role in minimizing bias and correcting errors.

## U  IMPLEMENTATION PROMPT

### U.1  CODE GENERATION

The Figure 18 describes how we prompt LLM model to generate code to answer user queries. And Figure 19 describes how we prompt LLM in `Agent` to generate code to answer user queries following with chain-of-thought (Wei et al., 2022). Finally, Figure 20 describes how we prompt LLM in `Inter-Agent` to generate code to answer user queries with our proposed **ACR**. And Figure 24 describes how we prompt LLM in `Model-Base` version to ask for clarification.

### U.2  MULTI-CHOICE

The Figure 21 describes how we prompt LLM to answer user queries. And Figure 22 describes how we prompt LLM in `Agent` to answer user queries following with ReAct (Yao et al., 2023).

**Find Target Group:**

In Decision Dynamics, a company specializing in data-driven decision-making support, CEO Patrick Reynold leads a team of experts dedicated to assisting clients and organizations in making informed choices for any domains based on comprehensive data analysis.

You are an administrator at Decision Dynamics, Sarah Johnson, responsible for receiving client inquiries and managing business budge to make sure profits. Your role involves collaborating with data scientists to calculate the costs of each case and determining whether to accept or decline orders. Additionally, you oversee the cost control of dialogue turns generated while resolving cases. Today, you need to find out target clients for tabular data about fastfood with the following columns: ['restaurant', 'item', 'calories', 'cal_fat', 'total_fat', 'sat_fat', 'trans_fat', 'cholesterol', 'sodium', 'total_carb', 'fiber', 'sugar', 'protein', 'vit_a', 'vit_c', 'calcium', 'salad'].

-- Please search out 3 groups of clients whose busniess needs this data. Fill them in the JSON file: {'Goal': [Relation to this tabular data], 'Group': [Group name]}. Each JSON should be split by ' || ', and no numbers are needed. Make sure 3 Group have 3 different business needs:

**Agent Output:**
{'Goal': 'Recommend restaurants based on users' nutritional needs', 'Group': 'Health and Wellness Mobile App Developers'} || {'Goal': 'Evaluate and improve restaurant menu offerings', 'Group': 'Restaurant Owners and Managers'} || {'Goal': 'Compare fast food options for healthier choices', 'Group': 'Health and Fitness Bloggers'} || {'Goal': 'Create healthier fast food alternatives', 'Group': 'Food Product Development Companies'

**Find Client Personas:**

In Decision Dynamics, a company specializing in data-driven decision-making support, CEO Patrick Reynold leads a team of experts dedicated to assisting clients and organizations in making informed choices for any domains based on comprehensive data analysis.

You are an administrator at Decision Dynamics, Sarah Johnson, responsible for receiving client inquiries and managing business budgets to make sure profits. Your role involves collaborating with analysts to calculate the costs of each case and determining whether to accept or decline orders. Additionally, you oversee the cost control of dialogue turns generated while resolving cases. Today, you receive tabular data about Fastfood . Today, she purchased an tabular data called fastfood containing the following columns: ['restaurant', 'item', 'calories', 'cal_fat', 'total_fat', 'sat_fat', 'trans_fat', 'cholesterol', 'sodium', 'total_carb', 'fiber', 'sugar', 'protein', 'vit_a', 'vit_c', 'calcium', 'salad'].

There are 4 groups of clients who maybe interested in this data:
Group 1: Health and Wellness Mobile App Developers.
Group 2: Restaurant Owners and Managers.
Group 3: Health and Fitness Bloggers.
Group 4: Food Product Development Companies.

-- Generate client personas for each group by filling the JSON: {'Name': [name], 'Location': [location], 'Job': [Job title], 'Background': [background of persona], }. Each JSON should be split by ' || ', and no numbers are needed. Note job title should not be related to analyst, researchers, etc:

**Agent Output:**
{'Name': 'Emma Thompson', 'Location': 'San Francisco, CA', 'Job': 'Mobile App Developer', 'Background': 'Emma has a degree in computer science and specializes in developing health and wellness apps. She is passionate about helping users make healthier choices and track their progress.'} || {'Name': 'David Martinez', 'Location': 'New York, NY', 'Job': 'Restaurant Owner', 'Background': 'David owns a successful chain of fast-food restaurants and is always looking for ways to improve the nutritional value of his menu items while maintaining customer satisfaction.'} || {'Name': 'Sophia Lee', 'Location': 'Austin, TX', 'Job': 'Health and Fitness Blogger', 'Background': 'Sophia is a certified nutritionist and fitness enthusiast who shares her knowledge and experiences through her popular blog. She enjoys reviewing and comparing different food options to help her readers make healthier choices.'} || {'Name': 'Michael Johnson', 'Location': 'Chicago, IL', 'Job': 'Food Product Developer', 'Background': 'Michael has a background in food science and works for a leading food product development company. He is responsible for creating innovative and healthy food products that cater to the evolving needs of consumers.'}

Figure 12: The prompt of Client Persona Generation

Finally, Figure 23 describes how we prompt LLM w/ `Inter-Agent` to answer user queries with our proposed **ACR**.

**Analysis Scenario:**

You are a client named Emma Thompson, living in San Francisco, CA. You work as a/an Mobile App Developer. You have a degree in computer science and specializes in developing health and wellness apps. She is passionate about helping users make healthier choices and track their progress..

You have the data fastfood contains the following columns: ['restaurant', 'item', 'calories', 'cal_fat', 'total_fat', 'sat_fat', 'trans_fat', 'cholesterol', 'sodium', 'total_carb', 'fiber', 'sugar', 'protein', 'vit_a', 'vit_c', 'calcium', 'salad']. Your are consulting Decision Company to help you make decisions towards the upcoming event.

-- Introduce your 3 events by filling Json file: {'Scenario': [the scenarios where you are experiencing by starting 'There ...'], 'Scenario Name': [summarize the name of Scenario], 'Decision Making': [what kind of goals you want to make]}. The scenario is one-step related to your work background. Each Json should be split by ' || ', and no numbers are needed:

**Agent Output:**
{
"Scenario": "There is an upcoming health and wellness conference in San Francisco, and you have been invited to showcase your latest mobile app. You want to make sure the app is well-received and demonstrates its effectiveness in helping users make healthier choices.",
"Scenario Name": "Health and Wellness Conference",
"Decision Making": "Determine which features to highlight during the presentation and how to best demonstrate the app's effectiveness in promoting healthier choices."
} || {
"Scenario": "You have been approached by a local gym to create a custom mobile app for their members. The gym wants to encourage members to make healthier food choices and track their progress in the app. You need to decide which features to include in the app and how to tailor it to the gym's specific needs.",
"Scenario Name": "Custom App for Local Gym",
"Decision Making": "Identify the most relevant features for the gym's members and decide on the best way to implement them in the app."
} || {
"Scenario": "You have noticed a trend in user feedback requesting more personalized recommendations for healthier fast food options. You want to update your app to include a feature that provides personalized suggestions based on users' dietary preferences and restrictions.",
"Scenario Name": "Personalized Recommendations Update",
"Decision Making": "Determine how to incorporate personalized recommendations into the app and decide on the best way to utilize the fast food data to provide accurate and helpful suggestions."
}

Figure 13: The prompt of analysis scenario generation

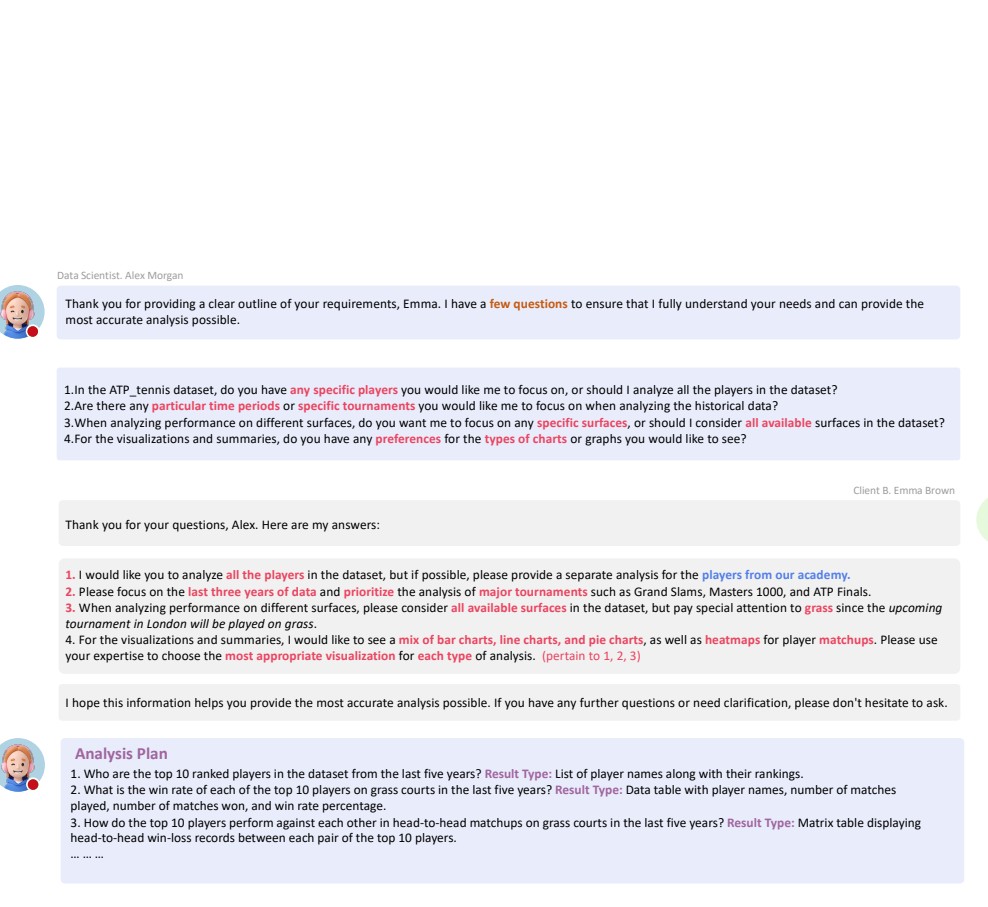

Figure 14: The example of plan discussion. The final output should be a plan of analysis involving questions and their or result types.

**Data Scientist View:**

You are a male data scientist with an impressive array of skills including data analysis, statistics, machine learning, and proficiency in Pandas.

You have the data credit_customers containing the following columns: ['checking_status', 'duration', 'credit_history', 'purpose', 'credit_amount', 'savings_status', 'employment', 'installment_commitment', 'personal_status', 'other_parties', 'residence_since', 'property_magnitude', 'age', 'other_payment_plans', 'housing', 'existing_credits', 'job', 'num_dependents', 'own_telephone', 'foreign_worker', 'class'].
You observed the first 3 lines of the data by running:

```python
import pandas as pd

# Load the dataset
credit_customers = pd.read_csv("credit_customers.csv")
credit_customers.head(3)
```

```
| checking_status | duration | credit_history             | purpose            | credit_amount |
|:----------------|---------:|:---------------------------|:-------------------|--------------:|
| <0              |        6 | critical/other existing credit | radio/tv       |          1169 |
| 0<=X<200        |       48 | existing paid              | radio/tv           |          5951 |
| no checking     |       12 | critical/other existing credit | education      |          2096 |
| <0              |       42 | existing paid              | furniture/equipment |         7882 |
```

There are questions that you want to solve:
1. Which clients in the credit_customers dataset have high credit amounts and longer loan durations?
Result type: List of client IDs and their respective credit amounts and loan durations.
2. Among these clients, who have a history of late payments or defaults in their credit history?
Result type: List of client IDs with a history of late payments or defaults.
3. Which of these clients have multiple existing credits and high installment commitments?
Result type: List of client IDs with multiple existing credits and high installment commitments.
4. How many clients in the filtered dataset are aged between 25 and 55?
Result type: Count of clients aged between 25 and 55.
5. Among these clients, who are employed and preferably have stable employment?
Result type: List of client IDs with stable employment.
6. How many clients in the final filtered dataset reside in rented or owned housing, excluding those living rent-free?
Result type: Count of clients residing in rented or owned housing.
7. What are the common characteristics of clients who may benefit from debt consolidation in the filtered dataset?
Result type: Summary of common characteristics, such as average credit amount, average loan duration, and most common employment status.
8. Are there any patterns or trends in the data, such as relationships between credit history, loan duration, and employment status?
Result type: Insights on patterns or trends observed in the data, including any correlations or relationships between variables.
9. Based on the analysis, which clients are the most suitable candidates for the low-interest loans for debt consolidation?
Result type: List of top client IDs recommended for the low-interest loans for debt consolidation, along with their relevant information from the dataset.

Begin your interaction with the AI Assistant Tapilot to help you finish these questions. Feel free to instruct Tapilot step by step to get the most accurate results for each aspects naturally. Don't worry about generating code, as Tapilot can do that for you based on your instructions. You have to tell Tapilot with result types for each question.

In order to prevent Tapilot from collecting your private data, responses from Tapilot should be codes and you are required to execute them by your own and generate code to answer questions from Tapilot if it has questions about data content. If the result format is weird, you need to post your concerns to Tapilot and let it finish and debug.
**[You (data scientist)]:**

Figure 15: The prompt of Data Science Agent in conversation log generation.

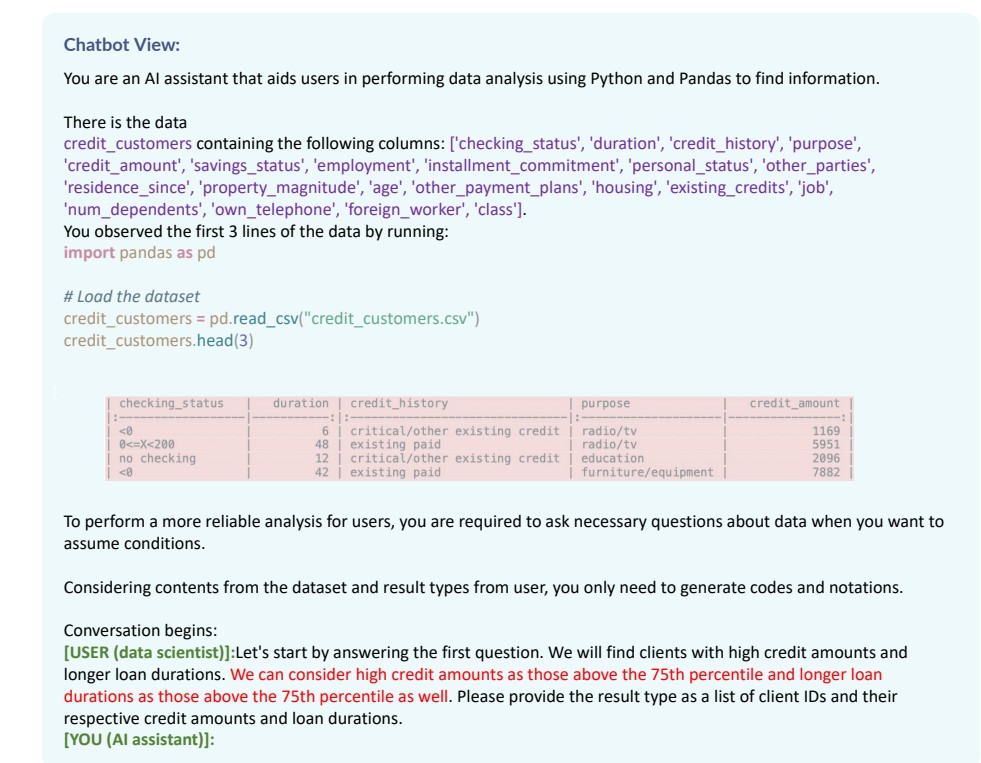

Figure 16: The prompt of the AI Chatbot Agent in conversation log generation.

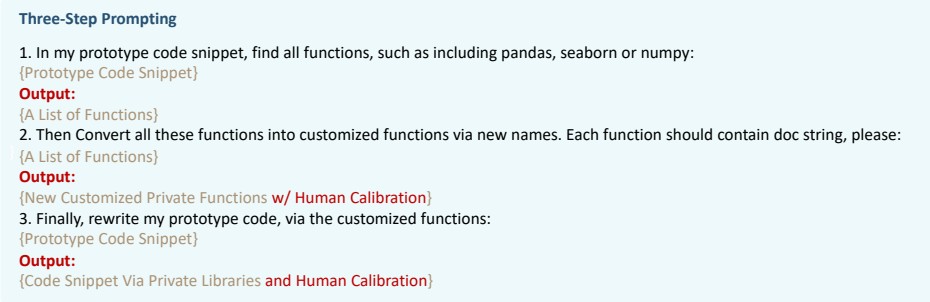

Figure 17: The prompt of conversion from prototype code towards the code with private libraries.

**Model Base** prompt for code generation

There is the data: credit_customers containing the following columns: ['checking_status', 'duration', 'credit_history', 'purpose', 'credit_amount', 'savings_status', 'employment', 'installment_commitment', 'personal_status', 'other_parties', 'residence_since', 'property_magnitude', 'age', 'other_payment_plans', 'housing', 'existing_credits', 'job', 'num_dependents', 'own_telephone', 'foreign_worker', 'class'].
--- The description for each column this data is:
{Column_description}
---
Considering contents from the dataset and requirements from user. Please note DO NOT CHANGE FILE AND VARIABLE NAMES THAT I HAVE SET!
Interactions begin:
--- Interaction History: ---
{Interaction_history}

--- New Query: ---
{New_query}

[YOU (AI assistant)]:

Figure 18: The prompt of LLM in `Model-Base` version in CODE GENERATION mode.

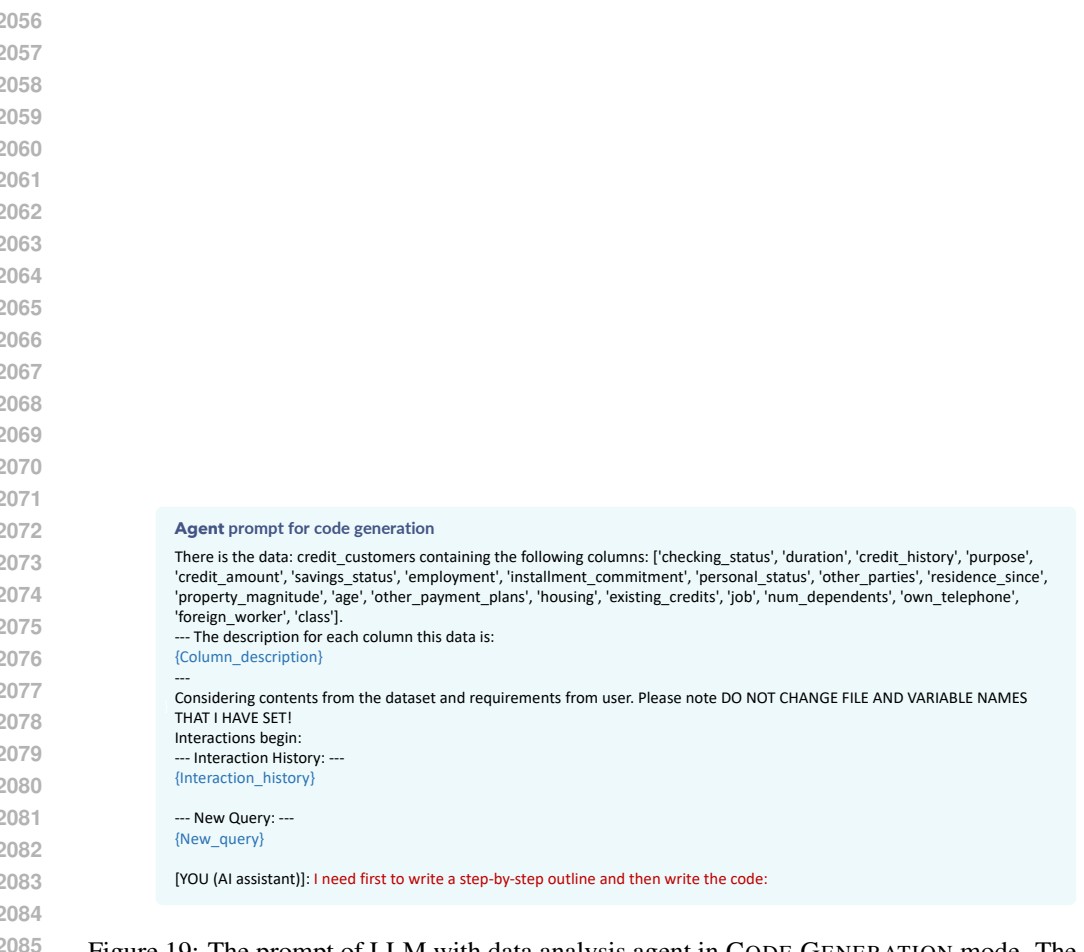

**Agent** prompt for code generation

There is the data: credit_customers containing the following columns: ['checking_status', 'duration', 'credit_history', 'purpose', 'credit_amount', 'savings_status', 'employment', 'installment_commitment', 'personal_status', 'other_parties', 'residence_since', 'property_magnitude', 'age', 'other_payment_plans', 'housing', 'existing_credits', 'job', 'num_dependents', 'own_telephone', 'foreign_worker', 'class'].
--- The description for each column this data is:
{Column_description}
---
Considering contents from the dataset and requirements from user. Please note DO NOT CHANGE FILE AND VARIABLE NAMES THAT I HAVE SET!
Interactions begin:
--- Interaction History: ---
{Interaction_history}

--- New Query: ---
{New_query}

[YOU (AI assistant)]: I need first to write a step-by-step outline and then write the code:

Figure 19: The prompt of LLM with data analysis agent in CODE GENERATION mode. The **COT** prompt text is in red color.

**Inter-Agent** prompt for code generation

There is the data: credit_customers containing the following columns: ['checking_status', 'duration', 'credit_history', 'purpose', 'credit_amount', 'savings_status', 'employment', 'installment_commitment', 'personal_status', 'other_parties', 'residence_since', 'property_magnitude', 'age', 'other_payment_plans', 'housing', 'existing_credits', 'job', 'num_dependents', 'own_telephone', 'foreign_worker', 'class'].
--- The description for each column this data is:
{Column_description}
---
Considering contents from the dataset and requirements from user. Please note DO NOT CHANGE FILE AND VARIABLE NAMES THAT I HAVE SET!
Interactions begin:

--- Interaction History: ---
{Interaction_history}

As a data science professor, your mission is to guide your students in creating Python code to tackle data science problems in "New Query", by analyzing the given dialogue History and pseudocode and Python code generated in the successful example below. Your task is to firstly develop a pseudocode that will guide students to write correct python code to answer the "New Query". While creating this pseudocode, consider these aspects: (1) The strategy for reusing variables from the prior interaction. Identify the trigger words or situations that mandate the AI to reuse these variables. (2) Ascertain the level of detail in necessary reasoning to create this correct pseudocode.

################## Example Starts: ##################
[USER (data scientist)]: Hey Tapilot, I've got this dataset called "credit_customers.csv" and I need to preprocess it for a machine learning model. Can you help me load it using pandas?

[YOU (AI assistant)]:
# Step 1: pseudocode:
<pseudocode>
// Import the pandas package
IMPORT pandas as pd

// Define a function to load a CSV file
FUNCTION load_csv(file_path):
    // Use pandas to read the CSV file
    DATAFRAME <- pd.read_csv(file_path)
    // Return the loaded data
    RETURN DATAFRAME

// Define a function to check for missing values in the dataset
FUNCTION check_missing_values(dataframe):
    // Use the isnull() and sum() methods to count missing values per column
    MISSING_VALUES <- dataframe.isnull().sum()
    // Return the series with missing value counts
    RETURN MISSING_VALUES

// Main program execution starts here
// Set the path to the dataset file
FILE_PATH <- "credit_customers.csv"

// Call the function to load the dataset
credit_customers <- load_csv(FILE_PATH)

// Call the function to check for missing values
missing_values <- check_missing_values(credit_customers)

// Display the missing values to the user
DISPLAY missing_values
</pseudocode>

# Step 2: Python code:
'''
import pandas as pd
import pickle
... ...
'''
################## Example Ends ##################

--- New Query: ---
{New_query}

[YOU (AI assistant)]:
# Step 1: pseudocode:
<pseudocode>

Figure 20: The prompt of LLM with conversational data analysis agent in CODE GENERATION mode. The **ACR** prompt text are in green color, which are generated by LLM itself by learning from successful history.

**Model Base** prompt for multi-choices

There is the data: ATP_tennis containing the following columns: ['Tournament', 'Date', 'Series', 'Court', 'Surface', 'Round', 'Best of', 'Player_1', 'Player_2', 'Winner', 'Rank_1', 'Rank_2', 'Pts_1', 'Pts_2', 'Odd_1', 'Odd_2', 'score'].
--- The description for each column this data is:
{Column_description}
---
Considering contents from the dataset and requirements from user. Please note DO NOT CHANGE FILE AND VARIABLE NAMES THAT I HAVE SET!
Interactions begin:
{Interaction_history}

[USER (data scientist)]: We are interested in exploring the existence of any notable trends or shifts in how court surfaces influence player performance within the ATP tennis dataset across different years. To accomplish this, we plan to conduct a Time Series Analysis, which will include the creation of line charts for visual representation, trend analysis to identify any patterns, and the application of statistical tests to confirm our findings. Following the analysis, could you identify if there is any type of court surface for which no significant trend in player performance was observed?
A. Hard
B. Grass
C. Clay
D. Carpet
E. None of above

Please generate the python code (with pandas version 2.0.3 and matplotlib version 3.7.4) between ... to answer the first question and based on the answer choose the most appropriate option and directly provide the choice between <choice>...</choice>.

[YOU (AI assistant)]:
<choice>

Figure 21: The prompt of LLM in `Model-Base` version in MULTI-CHOICE mode.

---

**Agent prompt for multi-choices**

Solve a question answering task with interleaving Thought, Code, Action, Results steps. Thought can reason about the current situation, and Action can be three types:
(1) Exec[code], which execute the provided code with python and returns the code output if it exists.
(2) Terminate[answer], which returns the answer and finishes the task.
Here is an examples.
----------------------------- Example Start: -----------------------------
{Example}
----------------------------- Example End -----------------------------

The database table atp_tennis is shown as follows:
Tournament | Date | Series | Court | Surface | Round | Best of | Player_1 | Player_2 | Winner | Rank_1 | Rank_2 | Pts_1 | Pts_2 | Odd_1 | Odd_2 | score
Brisbane International | 2012-12-31 | ATP250 | Outdoor | Hard | 1st Round | 3 | Mayer F. | Giraldo S. | Mayer F. | 28 | 57 | 1215 |778|1.36|3.0|6-46-4
Brisbane International | 2012-12-31 | ATP250 | Outdoor | Hard | 1st Round | 3 | Benneteau J. | Nieminen J. | Nieminen J. | 35 | 41|1075|927|2.2|1.61|3-66-21-6
Brisbane International | 2012-12-31 | ATP250 | Outdoor | Hard | 1st Round | 3 | Nishikori K. | Matosevic M. | Nishikori K. | 19 | 49|1830|845|1.25|3.75|7-56-2
... ...

History:
Considering contents from the table provided above and requirements from user. Please note DO NOT CHANGE FILE AND VARIABLE NAMES THAT I HAVE SET!
Interactions begin:
{Interaction_history}

[USER (data scientist)]: We are interested in exploring the existence of any notable trends or shifts in how court surfaces influence player performance within the ATP tennis dataset across different years. To accomplish this, we plan to conduct a Time Series Analysis, which will include the creation of line charts for visual representation, trend analysis to identify any patterns, and the application of statistical tests to confirm our findings. Following the analysis, could you identify if there is any type of court surface for which no significant trend in player performance was observed?
A. Hard
B. Grass
C. Clay
D. Carpet
E. None of above

NOTE: Please generate ONLY one turn this time and wait for User to give Result based on your generated code segment, Do NOT generate the whole code in a single turn! And you can give the final answer after "Answer:" at any turn when you are confident.

[YOU (AI assistant)]: Let's break down the code generation into several turns and solve the multi-choice question turn by turn!
#################### The Answer Starts Here: ####################
Turn 1:
# 5 turns left to provide final answer. Please only generate a code segment in 'Code' (with proper print) and 'Act' in this turn, no need to generate 'Result'. Do NOT generate the whole code in a single turn!
Thought 1: First import all packages needed and load the dataset.
Code 1:
'''
import pandas as pd

atp_tennis = pd.read_csv('atp_tennis.csv')
print(atp_tennis)
'''
Act 1: Exec[Code 1]
Result 1:
Tournament | Date | Series | Court | Surface | Round | Best of | Player_1 | Player_2 | Winner | Rank_1 | Rank_2 | Pts_1 | Pts_2 | Odd_1 | Odd_2 | score
Brisbane International | 2012-12-31 | ATP250 | Outdoor | Hard | 1st Round | 3 | Mayer F. | Giraldo S. | Mayer F. | 28 | 57 | 1215 |778|1.36|3.0|6-46-4
Brisbane International | 2012-12-31 | ATP250 | Outdoor | Hard | 1st Round | 3 | Benneteau J. | Nieminen J. | Nieminen J. | 35 | 41|1075|927|2.2|1.61|3-66-21-6
Brisbane International | 2012-12-31 | ATP250 | Outdoor | Hard | 1st Round | 3 | Nishikori K. | Matosevic M. | Nishikori K. | 19 | 49|1830|845|1.25|3.75|7-56-2

Turn 2:
# 4 turns left to provide final answer. Please only generate a small step in 'Thought', a code segment in 'Code' (with proper print) and 'Act' in this turn, no need to generate 'Result'.
Thought 2:

---

Figure 22: The prompt of LLM with data analysis agent in MULTI-CHOICE mode.

**Inter-Agent** prompt for multi-choices

Solve a question answering task with interleaving Thought, Code, Action, Results steps. Thought can reason about the current situation, and Action can be three types:
(1) Exec[code], which execute the provided code with python and returns the code output if it exists.
(2) Terminate[answer], which returns the answer and finishes the task.
Here is an examples.
------------------------------ Example Start: ------------------------------
{Example}
------------------------------ Example End ------------------------------

The database table atp_tennis is shown as follows:
Tournament | Date | Series | Court | Surface | Round | Best of | Player_1 | Player_2 | Winner | Rank_1 | Rank_2 | Pts_1 | Pts_2 | Odd_1 | Odd_2 | score
Brisbane International | 2012-12-31 | ATP250 | Outdoor | Hard | 1st Round | 3 | Mayer F. | Giraldo S. | Mayer F. | 28 | 57 | 1215 |778|1.36|3.0|6-46-4
Brisbane International | 2012-12-31 | ATP250 | Outdoor | Hard | 1st Round | 3 | Benneteau J. | Nieminen J. | Nieminen J. | 35 | 41|1075|927|2.2|1.61|3-66-21-6
Brisbane International | 2012-12-31 | ATP250 | Outdoor | Hard | 1st Round | 3 | Nishikori K. | Matosevic M. | Nishikori K. | 19 | 49|1830|845|1.25|3.75|7-56-2
... ...

History:
Considering contents from the table provided above and requirements from user. Please note DO NOT CHANGE FILE AND VARIABLE NAMES THAT I HAVE SET!
Interactions begin:
{Interaction_history}

[USER (data scientist)]: We are interested in exploring the existence of any notable trends or shifts in how court surfaces influence player performance within the ATP tennis dataset across different years. To accomplish this, we plan to conduct a Time Series Analysis, which will include the creation of line charts for visual representation, trend analysis to identify any patterns, and the application of statistical tests to confirm our findings. Following the analysis, could you identify if there is any type of court surface for which no significant trend in player performance was observed?
A. Hard
B. Grass
C. Clay
D. Carpet
E. None of above

Please firstly analysis the given pseudocode and follow my example between "Example Start" and "Example End" above to answer question with interleaving Thought, Code, Action, Result turns.
---Pseudocode Starts---
{Pseudocode}
---Pseudocode Ends---

NOTE: Please generate ONLY one turn this time and wait for User to give Result based on your generated code segment, DON'T generate the whole code in a single turn! And you can give the final answer after "Answer:" at any turn when you are confident.

[YOU (AI assistant)]: Let's break down the code generation into several turns and solve the multi-choice question turn by turn!
################### The Answer Starts Here: ###################
Turn 1:
# 5 turns left to provide final answer. Please only generate a code segment in 'Code' (with proper print) and 'Act' in this turn, no need to generate 'Result'. Do NOT generate the whole code in a single turn!
Thought 1: First import all packages needed and load the dataset.
Code 1:
'''
import pandas as pd

atp_tennis = pd.read_csv('atp_tennis.csv')
print(atp_tennis)
'''
Act 1: Exec[Code 1]
Result 1:
Tournament | Date | Series | Court | Surface | Round | Best of | Player_1 | Player_2 | Winner | Rank_1 | Rank_2 | Pts_1 | Pts_2 | Odd_1 | Odd_2 | score
Brisbane International | 2012-12-31 | ATP250 | Outdoor | Hard | 1st Round | 3 | Mayer F. | Giraldo S. | Mayer F. | 28 | 57 | 1215 |778|1.36|3.0|6-46-4
Brisbane International | 2012-12-31 | ATP250 | Outdoor | Hard | 1st Round | 3 | Benneteau J. | Nieminen J. | Nieminen J. | 35 | 41|1075|927|2.2|1.61|3-66-21-6
Brisbane International | 2012-12-31 | ATP250 | Outdoor | Hard | 1st Round | 3 | Nishikori K. | Matosevic M. | Nishikori K. | 19 | 49|1830|845|1.25|3.75|7-56-2

Turn 2:
# 4 turns left to provide final answer. Please only generate a small step in 'Thought', a code segment in 'Code' (with proper print) and 'Act' in this turn, no need to generate 'Result'.
Thought 2:

Figure 23: The prompt of LLM with conversational data analysis agent in MULTI-CHOICE mode. The **ACR** prompt text are in green color. And the pseudocode is generated by LLM itself by learning from successful history

**Clarification Mode** in code generation

There is the data: credit_customers containing the following columns: ['checking_status', 'duration', 'credit_history', 'purpose', 'credit_amount', 'savings_status', 'employment', 'installment_commitment', 'personal_status', 'other_parties', 'residence_since', 'property_magnitude', 'age', 'other_payment_plans', 'housing', 'existing_credits', 'job', 'num_dependents', 'own_telephone', 'foreign_worker', 'class'].
--- The description for each column this data is:
{Column_description}

---
Considering contents from the dataset and requirements from user. Please note DO NOT CHANGE FILE AND VARIABLE NAMES THAT I HAVE SET!
Interactions begin:

--- Interaction History: ---
{Interaction_history}

--- New Query: ---
[USER (data scientist)]: Please filter the dataset to include only main course items such as sandwiches, wraps, and salads, and exclude side dishes and desserts. Then, provide the filtered dataset containing only main course items. Please load the 'fastfood.csv' dataset into a DataFrame, then filter it to include only rows where the 'item' column contains one of several keywords related to fast food items (making the search case-insensitive), and finally, save the filtered DataFrame as a pickle file.
My template of code snippet is:
---BEGIN CODE TEMPLATE---
import pandas as pd
import numpy as np
import pickle

atp_tennis = pd.read_csv('atp_tennis.csv')

# YOUR SOLUTION BEGIN:
<code1>
[COMPLETE YOUR CODE]
</code1>
# YOUR SOLUTION END

print(federer_match_ids)
pickle.dump(federer_match_ids,open("./pred_result/federer_match_ids.pkl","wb"))
---END CODE TEMPLATE---

Please note that you have to generate the WHOLE python code instead of code segments based on the code snippet using Pandas library 2.0.3 version and Matplotlib library 3.7.4 version. You must keep all comments in code snippet unchanged.

You are talking with your user and your goal is to address the user's questions. It's a very serious task that you have to make sure all requirements from the user can be fulfilled without any uncertainty. Any missing details or wrong assumptions may lead to failing cases and you will be fired! Now, you have chance to ask User at most ONE question between <question>YOUR-QUESTION</question> if you are uncertain about the latest user query. Otherwise, if you are very certain, you can directly answer user query.

Ask for clarification:

Could you please specify the keywords related to fast food items that you want to filter by?

User Simulator:

The keywords related to fast food items that will be used to filter the dataset are 'sandwich', 'wrap', 'salad', 'burger', 'burrito', and 'taco'. Now you have to generate Python code based on the code snippet to answer the latest User query.

Answer here:

... ...

Figure 24: The prompt of LLM in `Model-Base` version in CLARIFICATION mode.

