# OpenReview forum: "Benchmarking Intelligent LLM Agents for Conversational Data Analysis"
_ICLR.cc/2025/Conference — Submitted to ICLR 2025_

### Official Review · Reviewer_KsMb · 2024-11-03

**Soundness:** 2
**Presentation:** 2
**Contribution:** 2
**Rating:** 5
**Confidence:** 4

**Summary:**

The paper introduced TAPILOT-CROSSING, a benchmark for evaluating LLM in conversational data analysis task. The dataset is generated with a multi-agent environment, DECISION COMPANY, with necessary human intervention. It used human evaluation to show the quality of the dataset, and evaluated various LLMs, with the proposed method Adaptive Conversation Reflection. The experiment result presented that current models performed poorly on the task, while ACR gave 44.5% boost from the base approach.

**Strengths:**

1. Synthetic conversation data generation incorporated with human in the loop data quality control.
2. Clear division of different conversation modes to reflect real-world scenarios.
3. Made use of multi-agent system to generate diverse and realistic dataset.
4. Proposed method (ACR) give huge boost than base LLM approach.

**Weaknesses:**

1. The abstract did not mention conversational data analysis requires tabular data input. In fact this is a very important input component of this paper. Please consider adding the related explanation.
2. The definition of conversational data analysis is unclear, and from the Related Work session, other papers' definition for this task also varies. For example, Wang et al., 2024 explained this task as one user turn and multiple cross-agent turns; Yan et al., 2023 saw this as multi user-agent turns. Based on the Task Formulation of this paper, sampled table content T is required in the setting. Since the definition of the task is not unified, this paper should explain clearly why a table is required, why the task is defined differently with others, and what distinct aspects of LLM do the task evaluating.
3. There are many terms in the paper used without definition or explanation, making the paper hard to understand. For example, requirements of client (Line 162), mainstream result types (Line 215), intents (line 246) are not mentioned before. Consider adding clear definition of the terms before using them.
4. The poor human evaluation result on the dataset before human calibration makes the overall DECISION COMPANY skeptical. The result represents that human is the main body of dataset construction, not the multi-agent system.
5. Baseline used in this paper is too weak. The paper used base LLM, CoT, and ReAct. For the code generation task, there are multiple recent advanced methods such as MetaGPT. Furthermore, the proposed method should be tested with various LLMs to show the generalizability.

**Questions:**

1. How do you ensure that the client persona is consistent with the table data (domain)?
2. What is the definition of 'reasonable scenarios' in human intervention for dataset? The detailed criteria will help this work reproducible.
3. For Line 168, What if the stereotype is not true on the dataset, making questions unanswerable?
4. For Line 193, how other choices of multiple-choice questions are made?
5. Could you add the number of turns in Data characteristics?

---

> ### Author Response · Authors · 2024-11-27
> **Request for Remaining Concerns**
>
> Thanks for your feedback and adjustment. We are glad that our responses can resolve your partial concerns. But could you kindly let us know where are remaining concerns or areas that you find still unsatisfactory? Could you make it more specific? We made considerable efforts in trying to address all the weaknesses and questions you raised, and we have revised the relevant parts of the paper and added experiments for the `multi-agent` baseline as per your recommendations.  We will make every effort to address remaining concerns or resolve misunderstandings thoroughly in the coming days. Thanks.

---

### Official Review · Reviewer_8PNW · 2024-11-04

**Soundness:** 3
**Presentation:** 4
**Contribution:** 3
**Rating:** 6
**Confidence:** 4

**Summary:**

The paper introduces TAPILOT-CROSSING, a novel benchmark for LLM evaluations in conversational data analysis, inspired by multi-agent LLM environments. Through rigorous human evaluations, the paper improves the reliability of human-AI approaches to constructing such conversational logs of data analyses in several action-focused dataset exploration scenarios. Also, the paper proposes Adaptive Conversation Reflection (ACR) to leverage previous conversation history to guide LLM agents to successful completion of the data analysis tasks.

**Strengths:**

- Very well-written and easy to understand with thoughtful explanations of details.
- Conducted a highly sophisticated design of dataset construction process with rigorous human validations, which strengthens the findings of the paper and the implications to future topics (e.g., beyond tabular data processing scenarios).
- They also conducted a qualitative analysis to identify error types across all models used, with a reasonable interpretation of the underlying reasons and patterns (as mentioned in the Appendix).

**Weaknesses:**

It appears that generating the 'logic' of the prior conversation trace and incorporating it into the next step of generation is not a novel approach to enhancing LLM reasoning in generative tasks. This ACR method closely resembles existing techniques, such as prompt chaining, ReAct, and self-reflection, in its methodological approach.

**Questions:**

- L154-160: in this phase human annotators only select a single scenario that sounds the most interesting. What is the agreement between the annotators in choosing the scenario during this phase?

---

### Official Review · Reviewer_MaWC · 2024-11-08

**Soundness:** 3
**Presentation:** 3
**Contribution:** 3
**Rating:** 5
**Confidence:** 4

**Summary:**

The paper introduces Tapilot-Crossing, a benchmark designed to evaluate large language models for conversational data analysis tasks. The benchmark contains 1,024 conversational interactions across four scenarios. A multi-agent environment was developed to create this benchmark, enabling automated and cost-effective data generation. The paper also proposes Adaptive Conversation Reflection (ACR), a self-reflective strategy to help LLMs learn from past interactions.

**Strengths:**

1. The paper provides an evaluation framework with diverse conversational scenarios, including scenarios requiring private library handling and complex conversational reasoning.

2. The paper presents an approach to scaling dataset generation cost-effectively, an essential aspect for building future benchmarks.

3. The paper evaluates multiple state-of-the-art LLMs and provides a granular analysis of their performance, highlighting challenges in conversational data analysis that underscore the need for improved LLM capabilities.

**Weaknesses:**

1. The main weakness is the potential over-reliance on simulated data: the exclusive reliance on simulated agent conversations might not fully capture the unpredictability and diversity of real-world human interactions in data analysis tasks.

2. While the paper introduces different scenarios, it lacks an in-depth justification for the selection of these specific conversational modes and how each addresses unique real-world challenges.

3. The results focus on improvements with ACR but offer limited exploration of failure cases and challenges within Tapilot-Crossing, such as common errors in multi-turn interactions.

**Questions:**

1. How were the conversational modes and action types specifically chosen, and were any other modes considered?

2. What types of errors were most commonly observed in scenarios involving private libraries, and how might future models address these errors?

3. Could human-in-the-loop interventions or feedback improve the realism of conversations, and if so, how would this influence the dataset’s construction costs?

---

### Meta-Review · Area_Chair_k93h · 2024-12-18

**Metareview:**

This paper introduces TAPILOT-Crossing dataset for benchmarking the conversational analysis capabilities of LLMs.  The dataset uses a collection of GPT agents playing different roles to generate conversations that are seeded from five Kaggle datasets.  The use of AI agents to generate conversations provides a cost and effort-effective method of dataset creation.  The authors were able to generate the dataset in under a month for less than $100.  One criticism is that only about a quarter of the dataset was ready to use with just the AI agents generation and the remainder had to be hand-edited by "two PhD students" who many or may not be paper authors.   The authors evaluate a sample of 500 conversations across a number of metrics using "10 experts" and show that the two PhD students edit the responses improved the scores.

This paper was considered a borderline contribution by reviewers with the main concern being the quality of the dataset.  Initial scores were 3,5 and 8 which after discussion became 5.6.5.  Due to the borderline nature of the assessment I also read the paper and had some questions about the quality of the dataset and the evaluation process.   I would have liked to see a more detailed analysis, maybe with a five point scale and a better description of the process - what were the rater instructions for the 10 expert raters?  They rated the conversation 0 or 1 for accept or reject.  If accept means that the conversation reasonably (perfectly?) meets the criteria I have a hard time believing that the score of 95% for Naturalness given the presented conversation in the appendix which does not seem like a natural human conversation.  As the authors point out in the weaknesses section that TAPILOT-CROSSING dataset assumes that all human-machine conversation history is clean and correct. However, in real-world scenarios, the conversation history is often not clean, and may contain noise or require multi-turn clarifications for a single question."

I am not convinced that this is a dataset that should be used for benchmarking at this point, but I want to appreciate that the authors put a lot of thought into how to efficiently generate a dataset.  I would like to see a better evaluation of the quality of the dataset.

**Additional Comments On Reviewer Discussion:**

The three remaining reviewers, all gave quality reviews.  Two of the reviewers actively engaged in discussion, one adjusted their rating up and the other down after reading authors responses.  The initial reviews were 3,5, and 8 and after discussion two of the reviewers changed their scores 3->5 and 8->6.

This paper unfortunately had one of the four reviewers drop out due to a personal emergency and I was unable to successfully replace them.

The conversation basically can be summarized as the paper has some good points but overall not quite good enough.

Personally I do not believe it was well evaluated, but it is an intersting way to generate datasets quickly. I am just ot sure how much I would trust it as a benchmark.

---

### Decision · Program_Chairs · 2025-01-22

Reject